# The role of P2Y12 in the kinetics of microglial self-renewal and maturation in the adult visual cortex in vivo

Monique S Mendes[1], Linh Le[1], Jason Atlas[1], Zachary Brehm[2], Antonio Ladron-de-Guevara[1,3], Evelyn Matei[1], Cassandra Lamantia[1], Matthew N McCall[2], Ania K Majewska[1,4]*

[1]Department of Neuroscience, University of Rochester Medical Center, Rochester, United States; [2]Department of Biostatistics, University of Rochester Medical Center, Rochester, United States; [3]Department of Biomedical Engineering, University of Rochester, Rochester, United States; [4]Center for Visual Science, University of Rochester, Rochester, United States

*For correspondence:
ania_majewska@urmc.rochester.edu

Competing interests: The authors declare that no competing interests exist.

**Abstract** Microglia are the brain's resident immune cells with a tremendous capacity to autonomously self-renew. Because microglial self-renewal has largely been studied using static tools, its mechanisms and kinetics are not well understood. Using chronic in vivo two-photon imaging in awake mice, we confirm that cortical microglia show limited turnover and migration under basal conditions. Following depletion, however, microglial repopulation is remarkably rapid and is sustained by the dynamic division of remaining microglia, in a manner that is largely independent of signaling through the P2Y12 receptor. Mathematical modeling of microglial division demonstrates that the observed division rates can account for the rapid repopulation observed in vivo. Additionally, newly born microglia resemble mature microglia within days of repopulation, although morphological maturation is different in newly born microglia in P2Y12 knock out mice. Our work suggests that microglia rapidly locally and that newly born microglia do not recapitulate the slow maturation seen in development but instead take on mature roles in the CNS.

## Introduction

Microglia are the brain's resident tissue macrophages (*Crotti et al., 2016*; *Ginhoux et al., 2010*) with a developmental origin that is distinct from other macrophages (*Ginhoux et al., 2010*). In mice, microglial progenitor cells derive from the yolk sac and populate the brain before the blood-brain barrier forms (*Ginhoux et al., 2010*; *Kierdorf et al., 2013*) slowly acquiring mature gene expression over two to three weeks (*Bennett et al., 2016*). In both the rodent and human brain, microglial numbers are maintained throughout adult life (*Réu et al., 2017*; *Füger et al., 2017*; *Hashimoto et al., 2013*), with no further contribution of peripheral macrophages to the microglial population in the absence of pathological changes (*Ajami et al., 2007*; *Askew et al., 2017*; *Bruttger et al., 2015*; *Elmore et al., 2015*; *Mildner et al., 2007*). Microglia are uniformly distributed in a distinct cellular grid throughout the brain parenchyma (*Nimmerjahn et al., 2005*; *Eyo et al., 2018*; *Hefendehl et al., 2014*) and maintain their territories with slow translocation on a timescale of days (*Eyo et al., 2018*).

Although microglia are long-lived cells, they do self-renew slowly under physiological conditions (*Füger et al., 2017*). To study microglial self-renewal, recent studies have pharmacologically and genetically depleted microglia and uncovered a remarkable capacity for rapid repopulation of the microglial niche without a contribution of infiltrating monocytes (*Ajami et al., 2007*; *Elmore et al., 2015*). Early studies in fixed sections from mice following depletion suggested that microglia rapidly

repopulate, possibly through a microglial progenitor population which can divide in large proliferative macroclusters from which newly born microglia migrate throughout the brain (*Bruttger et al., 2015*; *Elmore et al., 2014*). More recent studies have shown that surviving microglia divide and transiently express nestin; however, none of the repopulated microglia derive from nestin progenitors (*Huang et al., 2018a*; *Huang et al., 2018b*; *Zhan et al., 2019*). Similar to repopulation in the cortex, microglial repopulation in the retina is driven by remaining microglia; however, proliferation occurs in the central retina, suggesting a specific site of microglial generation in the eye (*Huang et al., 2018b*; *Zhang et al., 2018*). In the retina, repopulated microglia eventually adopted the dynamic features of endogenous microglia before depletion (*Zhang et al., 2018*), and repopulated cortical microglia also have expression patterns similar to those of endogenous microglia 30 days after repopulation (*Bruttger et al., 2015*; *Huang et al., 2018b*; *Zhan et al., 2019*). Genomic analysis of newly born cortical microglia shows that new microglia within days of repopulation recapitulate developmental expression profiles (*Elmore et al., 2015*; *Zhan et al., 2019*), suggesting that they may undergo a stepwise maturation after repopulation that resembles developmental maturation.

Microglia are highly dynamic, and understanding their behavior requires an equally dynamic approach that monitors their characteristics chronically over time in their native milieu. Here, we characterized microglial ontogeny and maturation in the adult visual cortex using time-lapse imaging in vivo in awake young adult mice after microglial depletion. We show that microglial self-renewal is slow under basal conditions, with microglia maintaining their territories and showing little movement, loss, or proliferation. In agreement with previous studies, we show that, following depletion, newly born microglia rapidly repopulate the brain and acquire equal cell-to-cell spacing reminiscent of baseline conditions. Self-renewal is driven locally by residual microglia, which have a capacity for fast and continued self-division. To determine whether other mechanisms such as migration from sites outside of the imaging area contributed to the increase in microglia numbers, we mathematically modeled microglial self-renewal in vivo and showed that the observed division rates could account for the rapid repopulation. Finally, we showed that newly born microglia quickly acquire ramified morphologies and dynamic surveillance capabilities in response to focal injury, suggesting that microglial functional maturation is remarkably rapid in the adult brain and does not recapitulate developmental features. To dissect the underlying mechanisms responsible for our observations of microglial repopulation, we considered signaling pathways in microglia associated with migration, microglial motility, and maturation. While there are many receptors expressed by microglia that respond to changes in their environment, we focused on the P2Y12 receptor. P2Y12 is highly expressed exclusively in microglia in the brain (*Bennett et al., 2016*) and regulates microglial translocation under physiological conditions in vivo (*Haynes et al., 2006*; *Eyo et al., 2014*). Neither microglial repopulation nor the maturation of newly-born microglia was altered by P2Y12 loss. These findings suggest that the microglia landscape following depletion is restored through a rapid division of remaining microglia, local migration to fill the microglia niche, and fast acquisition of mature characteristics soon after repopulation without the need for P2Y12 signaling.

## Results

### *In vivo* imaging of microglia shows limited migration and turnover with preserved territories in the physiological brain

To track microglial turnover in vivo under basal conditions, we imaged microglia daily in the same awake young adult mice using the microglial-labeled CX3CR1-GFP transgenic mouse line (*Jung et al., 2000*) and a chronic cranial window preparation. Because a growing body of literature suggests that microglia behave differently under anesthesia (*Stowell et al., 2019a*; *Li et al., 2012*; *Sun et al., 2019*), we wanted to capture the dynamics of microglia in the absence of anesthesia to avoid potentially inducing long-term alteration in microglial movement and turnover during repopulation. Microglia in the same cortical area could easily be imaged and tracked from day to day (*Figure 1A* and *Video 1*) without overt changes in microglial morphology, which could indicate immune activation over time. Microglia numbers were stable over 14 consecutive days, and individual microglia could be re-identified in the same location daily over this period, suggesting limited local migration and maintenance of microglial territories during this time (*Figure 1A*). Quantitative three-dimensional nearest neighbor (NN) analysis revealed that microglia generally maintain their

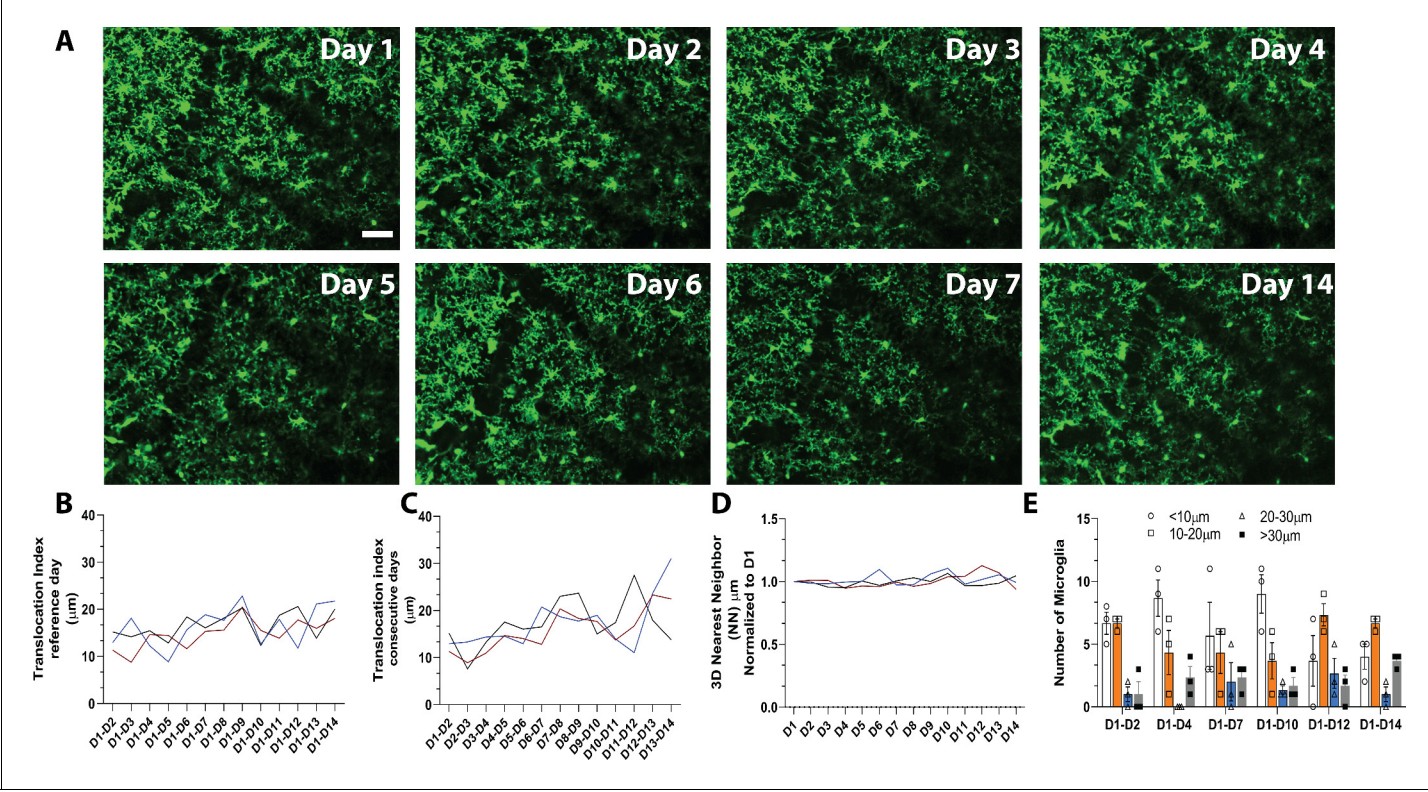

**Figure 1.** *In vivo* imaging of microglia shows limited migration and turnover in the physiological brain. (A) A field of microglia in an awake mouse imaged over 14 consecutive days. The dark diagonal lines in the center and top right of the image are blood vessels that remain structurally stable and can be used as landmarks to identify the same location for chronic imaging. Several microglia are identified with the same-colored circles at different time points to show the stability of their somas. (B) Nearest neighbor quantification in 3D demonstrates the distribution of neighboring microglial cells over consecutive days. (C) The translocation index, which captured the average displacement of microglia over time, was ~15 μm when consecutive imaging sessions were compared. (D) The translocation index increased when D1 is compared to imaging carried out later (D2–D14). (E) Microglia translocation between D1-D2, D1-D4, D1-D7, D1-D10, D1-D12, and D1-D14. On average, the majority of microglia remained within ~10 μm away (white bars, circles) from their original location. The number of microglia that moved within their domain (10–30 μm; orange bars, squares), (20–30 μm; blue bars, triangles) or translocated a further distance (>30 μm; gray bars, filled circles) stayed relatively constant with increasing interval between imaging sessions (n=3, 30–40 μm stacks, 13–17 microglia per mouse). Scale bar, 50 μm. *Figure 1—source data 1*: Source data for microglia turnover and migration in the physiological brain.

The online version of this article includes the following source data and figure supplement(s) for figure 1:

**Source data 1.** Source data for microglia turnover and migration in the physiological brain.
**Source data 2.** Source data for CSF1R inhibition in the adult brain.
**Source data 3.** Source data for the Cytokine panel conducted.
**Figure supplement 1.** CSF1R inhibitor consistently eliminates microglia from the adult brain.
**Figure supplement 2.** No change in inflammatory milieu following microglial depletion.

territories, whereby each microglia soma lies at ~30 μm from its nearest neighboring microglia. This NN distance remained similar on subsequent days and between animals (*Figure 1B*, n=3, 16–17 microglia per mouse, 30–40 μm stacks).

Microglial distribution over time was assessed with a custom algorithm that compared the location of microglia in the same field of view over the 14-day imaging period. To measure microglial movement, we defined a 'translocation index,' which was an average distance between the location of each microglia on day 1 and the location of the nearest microglia on an nth day (n=3, 16–17 microglia per mouse, 30–40 μm stacks). On average, microglia shifted on the order of 15 μm from day to day (*Figure 1C*). This number increased as the two imaging sessions were further apart in time (*Figure 1D*), possibly due to tissue distortion over time. In general, most microglia did not move by more than 10 μm in either short (1–2 day), (1–4 day), (1–7 day) and long (1–10 day), (1–12 day), (1–14 day) comparisons. The proportion of microglia that moved more than 30 μm (the

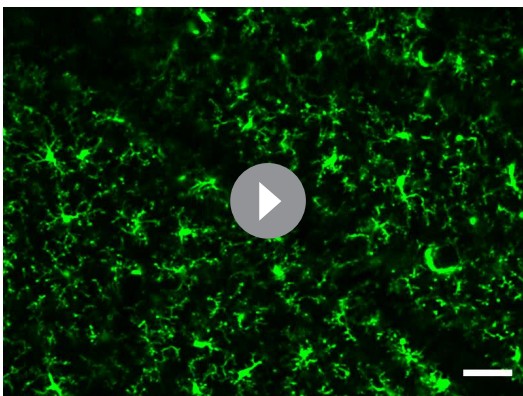

**Video 1.** Example in vivo two-photon imaging in awake adult mice. Two-photon in vivo images were obtained in CX3CR1-GFP awake mice following three consecutive days of training and habituation to the setup with a 1.5x zoom. This video shows a 100-slice z-stack starting at the pial surface and going deeper into the brain in a control awake animal. Microglia are evenly spaced and maintain this organization over many imaging sessions under control conditions. (Microglia are green). Scale bar, 50 μm.
https://elifesciences.org/articles/61173#video1

territory of single microglia) was small in both short and long-term comparisons (*Figure 1D–E*, n=3, 16–17 microglia per mouse, 30–40 μm stacks). Overall, this analysis suggests that microglia are stable under basal conditions and show limited movement within their territory over 14 days.

## Microglia rapidly replenish after partial depletion in the visual cortex

Microglial homeostasis is tightly regulated under basal conditions. Therefore, to explore microglia self-renewal in the visual cortex, we used an established paradigm that partially depletes microglia using PLX5622 (PLX), a Colony Stimulating Factor 1 Receptor (CSF1R/c-kit/Flt3) inhibitor (*Elmore et al., 2014*; *Dagher et al., 2015*; *Najafi et al., 2018*). With the introduction of PLX for 7 days, microglia numbers in the visual cortex decreased by 75% (*Figure 2A,B*; *Figure 2—figure supplement 1*), and the 3D NN number increased to 200 μm reflecting the decreased density and therefore increased distance between remaining microglia (*Figure 2C*). We did not observe changes in cytokine production or astrocyte morphology following depletion (*Figure 1—figure supplement 2* and *Figure 2—figure supplement 1*).

After discontinuing PLX treatment, we imaged microglial restoration daily in the same mouse to capture the dynamics of this process (*Figure 2A*). Newly born microglia rapidly repopulated the visual cortex, and microglia numbers were almost completely restored after only 3 days of repopulation. In addition, the newly generated microglia surpassed baseline numbers after 5–7 days of repopulation (*Figure 2B*). In concert, the 3D NN distance returned to control levels after 3 days of repopulation and remained relatively stable until 30 days of repopulation, the last time point examined, at which time the number of microglia was similar to that of control microglia numbers before PLX treatment (*Figure 2C* and *Video 2*). These results suggest that microglial proliferation occurs very rapidly over a 24-hr period starting at ~2 days after cessation of PLX treatment and that microglia rapidly regain their territories within the visual cortex, maintaining their equal spacing and numbers after repopulation is complete.

PLX treatment in P2Y12-KO/CX3CR1-GFP (which will be abbreviated to P2Y12-KO henceforth) mice caused a depletion of microglia of ~70% (*Figure 2D*), with a concomitant increase in 3D NN (*Figure 2E*). The change in NN distance was not as profound as in CX3CR1-GFP mice, possibly because these animals had, on average, smaller magnitudes of depletion, although depletion was highly variable across all animals examined irrespective of genotype (*Figure 2F*). While microglia repopulated in a temporal pattern similar to CX3CR1-GFP mice after cessation of the inhibitor (*Figure 2D,E*), there appeared to be a slight delay in early repopulation (*Figure 2G*). However, this delay resolved quickly, and on day 5 and beyond, microglial repopulation matched CX3CR1-GFP controls (*Figure 2H,I* and *Figure 2—figure supplement 2* and *Figure 2—figure supplement 3*). This suggests that P2Y12 signaling plays a minor role in repopulation dynamics.

## Residual microglia are capable of rapid division to generate new microglia and repopulate the cortex

Microglial self-renewal may be driven by local residual microglia that remain following depletion, although the dynamics of this process are poorly understood. It is unclear whether specific subpopulations of remaining microglia are responsible for the division, or all remaining cells have the capacity

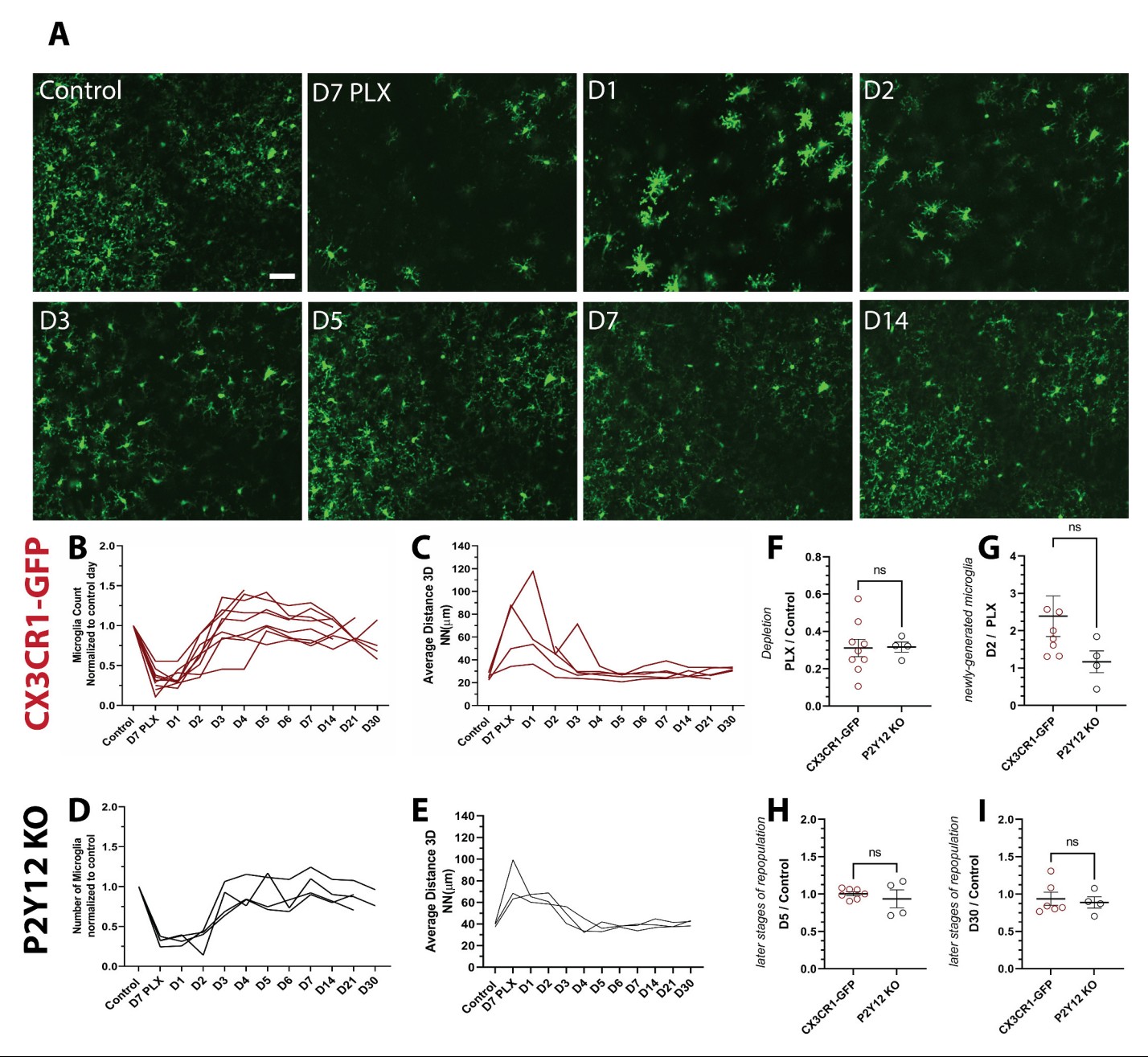

**Figure 2.** Microglia rapidly repopulate the visual cortex after partial depletion. (A) A field of microglia during depletion and repopulation imaged in vivo in the same awake mouse. (B) The number of microglia (normalized to control day for each animal) during depletion (PLX) and with repopulation (day 1 day 30). Each line represents an individual animal (n=10, 70–80 microglia per mouse). (C) 3D nearest-neighbor quantification showed a large increase during depletion and the early stages of repopulation before returning to control numbers (n=4, 5–80 microglia per mouse; a subset of mice from (C) that could be imaged throughout the control, depletion, and repopulation time points were used for this analysis). (D) Depletion and repopulation dynamics were similar in the absence of P2Y12 (n=four animals, 5–80 microglia per mouse). (E) 3D nearest neighbor analysis shows similar changes in microglial distribution during repopulation in the absence of P2Y12 (n=3 animals 5-80 microglia per mouse). (F) The ratio of microglia numbers observed on D7 PLX to control. (G) Repopulation was slightly delayed in P2Y12-KO mice as compared to WT, as the change in microglial numbers from depletion (PLX) to day 2 of repopulation was significantly smaller in the absence of P2Y12. (H) By day 5 of repopulation, the change in microglial numbers had normalized between WT and P2Y12-KO mice. (I) Microglial numbers never fully recovered to control conditions in either WT or P2Y12-KO mice (F-I, T-test ns, n=4, 5–80 microglia per mouse). Scale bar, 50 µm. *Figure 2—source data 1*: Source data of microglial repopulation after partial depletion.

The online version of this article includes the following source data and figure supplement(s) for figure 2:

**Source data 1.** Source data of microglial repopulation after partial depletion.

*Figure 2 continued on next page*

*Figure 2 continued*

**Source data 2.** Source data for the GFAP expression following microglial depletion.
**Figure supplement 1.** GFAP expression is unchanged following microglial depletion.
**Figure supplement 2.** Comparison of microglial number over the course of repopulation in the presence and absence of P2Y12.
**Figure supplement 3.** Comparison of nearest neighbor distance over the course of repopulation in the presence and absence of P2Y12.

to divide; whether repopulation is local or originates in a specific brain region and is coupled to large scale microglial migration; and whether repopulating cells do so by single-cell division or through an intermediate multinucleate body that gives rise to multiple new microglia. During the repopulation phase, we observed occasional splitting of existing microglia into two daughter cells (*Figure 3*), which has been reported previously (*Füger et al., 2017*; *Tay et al., 2017*). Both new cells were present in subsequent imaging sessions, indicating that these cells persist and become integrated with the microglial network (*Figure 3A*).

The characteristic morphologies of microglia in the process of division allowed us to identify 'potential' proliferating cells, which we refer to as doublet somas. To determine whether these doublet microglia had distinct morphological characteristics, we analyzed the morphology of microglia in fixed sections from control mice and mice whose microglia repopulated for two days after depletion (peak proliferation; *Figure 3F–I*). Doublet microglia during repopulation, had larger and more elongated somas than either control microglia or singlet microglia during repopulation although all these parameters overlapped in the three microglia populations (*Figure 3I–K*). A combination of soma perimeter and soma elongation effectively separated out doublet microglia from singlet microglia after repopulation and from microglia in control animals, showing that doublet microglia have a distinct morphological signature. It is notable that this analysis suggests that singlet microglia show a larger morphological diversity than control microglia, suggesting that newly born microglia themselves may have an altered morphology. Interestingly, Galectin-3 expression was increased in repopulating microglia, although its expression was not statistically significantly different between doublet and singlet microglia (*Figure 3—figure supplement 1*), recapitulating previous results showing that proliferating microglia can be Galectin-3 positive or negative (*Zhan et al., 2020*). Additionally, only a subset of doublet cells was Galectin-3 positive.

We quantified both the total number of doublets and the percentage of all microglia that were doublets during our imaging of depletion and repopulation and found that both these measures were increased in the repopulation phase (*Figure 3B,C*). Similar dynamics of doublet microglia was observed in P2Y12-KO, again suggesting that P2Y12 signaling did not play a large role in regulating microglial division (*Figure 3D, E*). While these dividing cells could contribute to the rapid increase in microglial numbers observed, three pieces of evidence suggested that other mechanisms may be important. First, in our model of depletion, the number of doublet cells per field increased around day 3 to day 5 of repopulation, but microglia had largely repopulated the brain by day 3 after cessation of PLX. Second, the time scale of the division of each doublet was slow, with microglia remaining in the doublet state for days before dividing, making it unlikely that these cells could create the large increases seen in microglial numbers within 3 days (*Figures 2B* and *3B,C*). Third, the percentage of microglia that were in a doublet state was low even during peak repopulation,

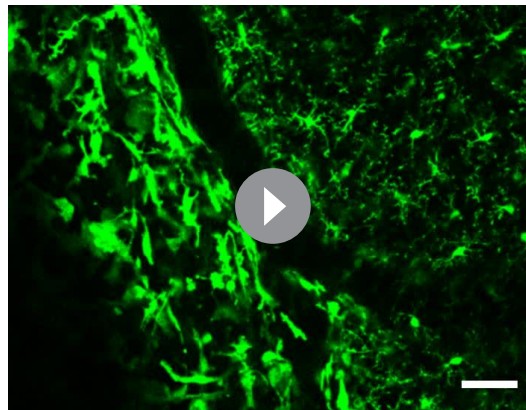

**Video 2.** Example in vivo two-photon imaging in awake adult mice during depletion and repopulation. Two-photon in vivo images were obtained in CX3CR1-GFP awake mice following three consecutive days of training and habituation to the setup with a 1.5x zoom. This video shows a 100-slice z-stack in the same region during depletion (2D, 4D, 6D, and 7D PLX) and repopulation (D1-D7, D14, D21, and D30). Scale bar, 50 µm.
https://elifesciences.org/articles/61173#video2

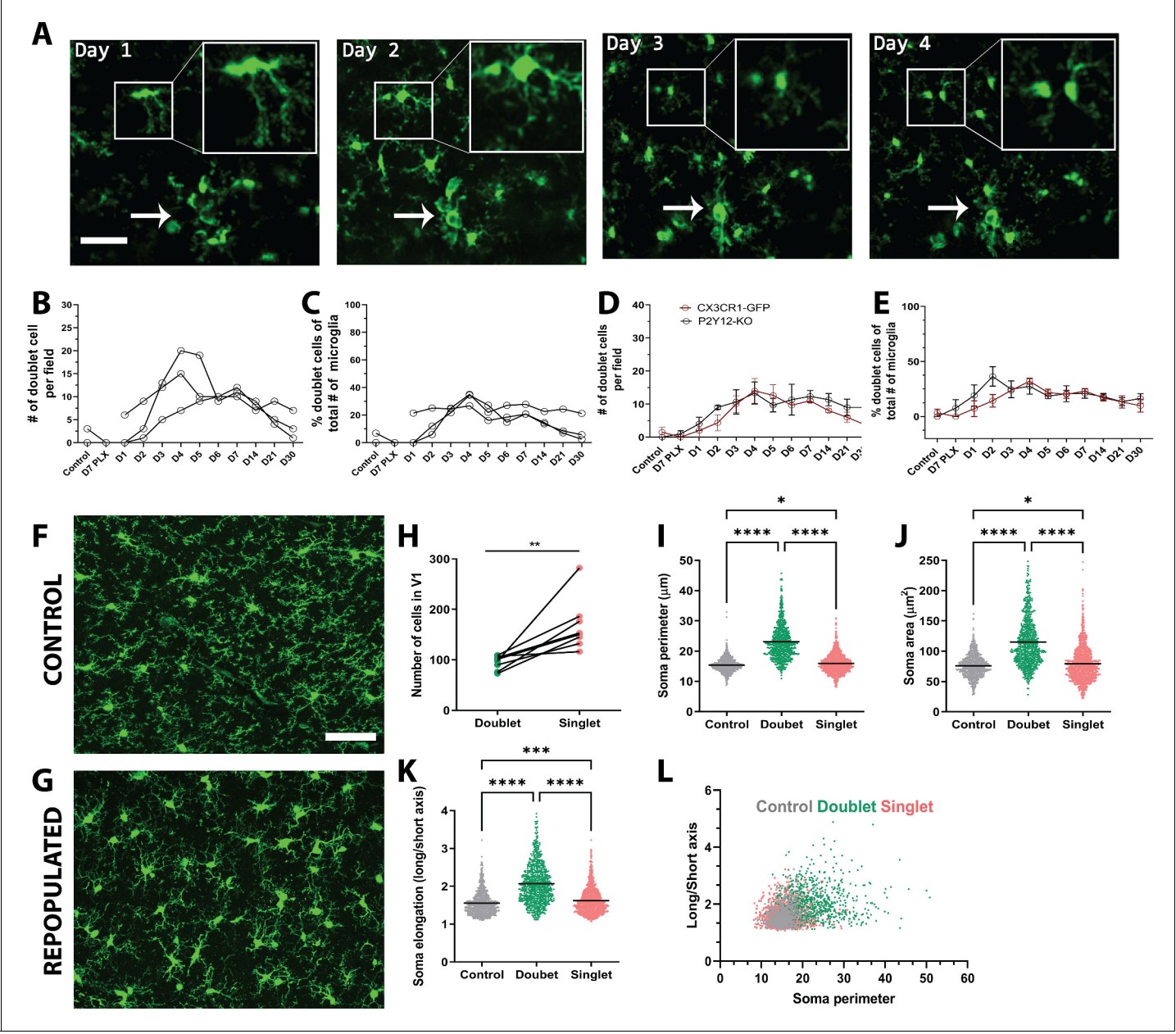

**Figure 3.** Microglia self-renewal via residual cells in the visual cortex. (**A**) Representative in vivo two-photon imaging of microglial division in CX3CR1-GFP awake mice. Inset is a magnified image of the dividing doublet microglia, which appears as an elongated cell body on day 1 and divides into two cells on day 2. The two microglia then migrated away from one another over the subsequent 2 days. The arrow indicates a cluster of microglia that may represent a multinucleate body but did not generate new microglia during the imaging period. (**B**) The number of doublet cells per field increased at day 3 day 5 of repopulation (n=3–4, 50–150 microglia per mouse). (**C**) Doublets made up close to 40% of the total number of microglia at day 4 of repopulation (n=3–4, 50–150 microglia per mouse). The number (**D**) and percentage (**E**) of microglial doublets over time were similar in P2Y12-KO mice as compared to WT (n=three animals per group). (**F–G**) Representative confocal images of microglia from control CX3CR1-GFP mice (**F**) or mice 2 days after cessation of PLX (peak repopulation) (**G**) (n=four animals per group). Examples of singlets and doublets are shown with blue and red arrows, respectively. (**H**) Doublet cells made up ~40% of the total number of microglia quantified during the peak of repopulation. Doublets present after 2 days of repopulation showed an increase in both soma perimeter (**I**), area (**J**) and elongation (as defined by the ratio of the lengths of the long to the short axis; **K**), as compared with singlets and control microglia. (**L**) Soma elongation plotted against soma perimeter shows separation of the defined morphological states. While, control and singlet microglia overlap, doublet microglia are relatively distinct. (*p<0.05, **p<0.01, ***p<0.001, ****p<0.0001, T-test, paired (**H**); One-way ANOVA, Dunnett post-hoc test (**I–K**)). Graphs show mean ± s.e.m. Each line represents an animal (**B–E**), points represent individual animals (**H**), and each dot represents a cell (**I–L**). Scale bar, 50 μm. *Figure 3—source data 1*: Source data for microglial repopulation.

*Figure 3 continued on next page*

*Figure 3 continued*

The online version of this article includes the following source data and figure supplement(s) for figure 3:

**Source data 1.** Source data for microglial repopulation.
**Source data 2.** Galectin three expression in microglia after repopulation.
**Figure supplement 1.** Galectin three expression in doublet and singlet microglia.

and this level of doublets in the microglial population was maintained long after microglial repopulation was complete (*Figures 2B* and *3B,C*).

To determine whether microglia are capable of more rapid division that could account for the rapid increase in microglial numbers seen 2–3 days following cessation of PLX treatment, we imaged awake young adult mice every 4 hr for 24 hr during the most dynamic time of repopulation (Day 2 Day 3; *Figure 4*). Unlike what we observed during once per day imaging sessions (*Figure 2* and *Figure 3*), a microglial division could be remarkably rapid where doublets separated into two daughter cells in as little as 4 hr (*Figure 4B*). In an analysis of a subset of divisions where cells could be clearly tracked throughout the 24-hr period (22 total from three animals), we found that 91% of new cells were associated with a characteristic increase and elongation of the cell soma of the original cell before it divided into two microglia. Original microglia often had extensive processes, which were maintained during the division and generation of new microglia (95%), and newly generated microglia frequently had a ramified microglia arbor on the same day that division was complete (82%). Cells generated from these divisions persisted over time, moving away from one another to establish their own territories (100% of cells; *Figure 4*).

While division could be very rapid, the rates of microglia division were not uniform, and some cells divided slower within 8 or 12 hr or remained in the doublet state for the duration of imaging (*Figure 4A*). Notably, in some cases, we observed cells dividing twice (*Figure 4C,D*). The first division occurred rapidly, typically within 4 hr, and the cells migrated away from one another. These cells then adopted a doublet morphology and divided again in less than 8 hr (*Figure 4C,D*). The rate of microglial division did not appear to be spatially regulated as microglia in close proximity divided at different times (*Figure 4D*). Microglial numbers roughly doubled over this imaging period in our animals (*Figure 4E*), and 3D nearest-neighbor numbers fell as microglia were generated (*Figure 4F*). The number of doublet cells also increased but tended to plateau around 12 hr into the imaging period (*Figure 4G*). In addition, the number of rapid divisions, as well as secondary divisions of newly born microglia also increased (*Figure 4H*). To determine how mobile microglia were during the peak of repopulation, we tracked a small, identified subset of microglia in three of these animals. Microglia translocation index during 4 hr intervals over 24 hr was ~8 μm (*Figure 4I*). Overall, most microglia did not move by more than 10–20 μm over the 24 hr. The proportion of microglia that moved 20–30 μm was small during this time course (*Figure 4J*). More in-depth analysis of one of the animals to track and classify microglia showed that only 7% of microglia during this time did not divide, and 5% of these adopted a doublet morphology that persisted throughout the imaging session, suggesting they may divide at a later time. Of new cells, 20% were born by rapid division, while only 2% of microglia were born of secondary divisions in this mouse (*Figure 4K*). Overall, this points to a remarkable capacity of the majority of the remaining microglia to divide rapidly and suggests that this rapid division may explain the fast repopulation of the cortex after depletion.

## Mathematical modeling of microglial repopulation kinetics in the adult visual cortex

To determine whether local remaining microglia could repopulate the cortex through rapid division, we created a mathematical model (*Figure 5*; *Figure 5—figure supplement 1*) which used the division rates of newly-born microglia quantified in the 24 hr imaging experiment (*Figure 4*) to determine whether these rapid divisions could account for the increases in microglial numbers seen when repopulation was imaged daily (*Figure 2*). The model considered three parameters: (*Crotti et al., 2016*) the number of cells in the population on day 2 of the daily imaging paradigm (for each individual animal imaged in *Figure 2*), (*Ginhoux et al., 2010*) the probability that a cell will be eligible for division (we tested using proportions ranging from 10 to 100% in increments of 10), and (*Kierdorf et al., 2013*) the observed division rates of the cells during the peak of repopulation

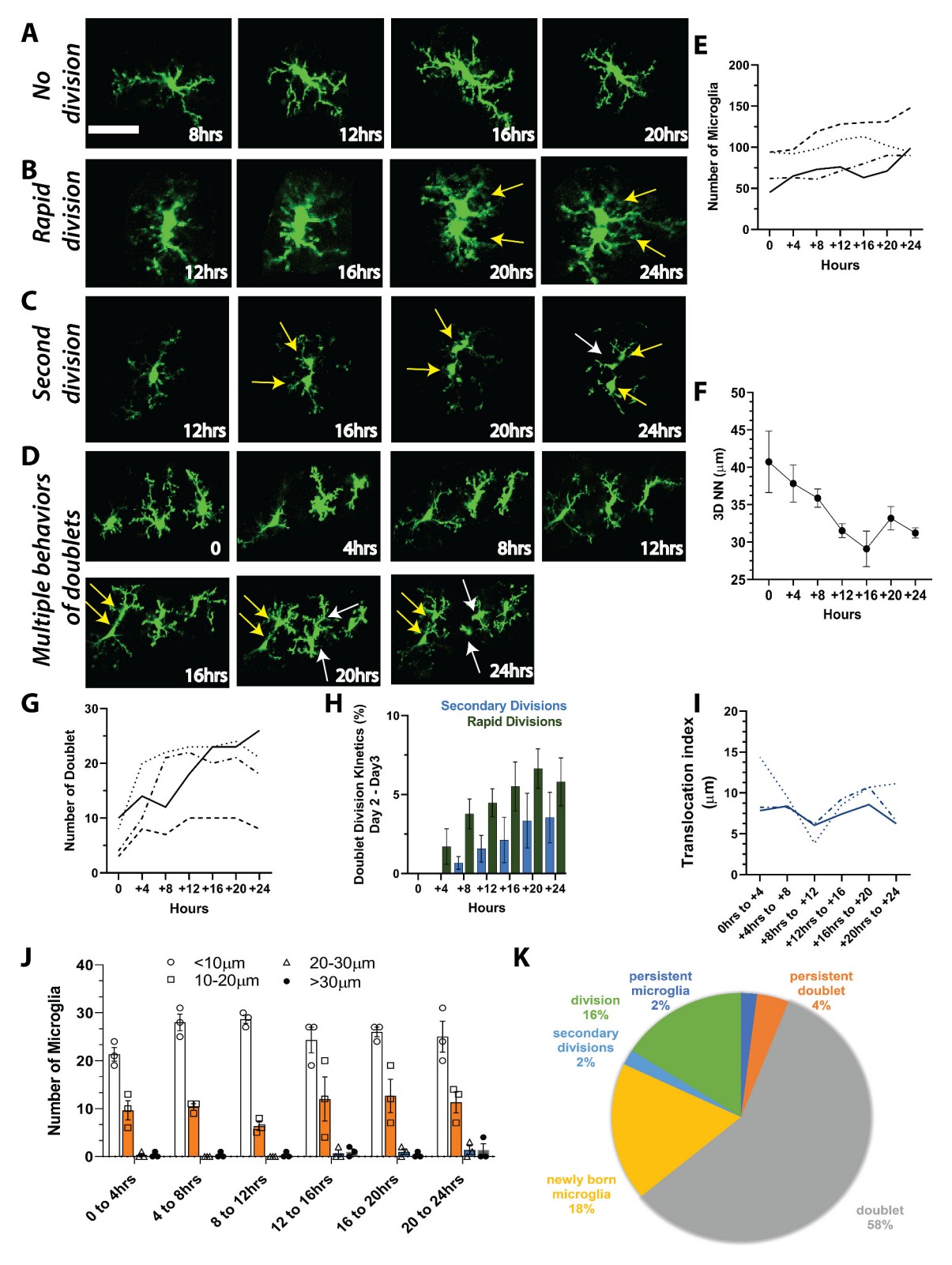

**Figure 4.** Microglia repopulation in the visual cortex demonstrates that cell division can account for the fast repopulation observed *in vivo*. We imaged CX3CR1-GFP mice every 4 hr for 24 hr during the peak of repopulation (day 2 day 3). We observed a range of behaviors of existing microglia: (**A**) no division, (**B**) rapid division on the time scale of 4–8 hr once the doublet appeared, and (**C**) secondary division, where newly divided microglia underwent another division. (**D**) Microglial divisions are not spatially regulated. Three microglia in the field of view are divided at different times over the course of

*Figure 4 continued on next page*

*Figure 4 continued*

24 hr. (**E**) The number of microglia during the 24-hr period in the four mice that were imaged. (**F**) Microglial 3D nearest neighbor distance decreased with time as more microglia were added to the population (n=3). (**G**) The number of doublets increased during the 24-hr period (n=4). (**H**) Rapid divisions made up the larger proportion of divisions at each imaging time point compared to secondary divisions (n=4). (**I**) We tracked identified microglia over time in a small field of view in a subset of animals. The translocation index, which captured the average displacement of microglia over time, was ~8 µm when consecutive imaging sessions were compared (n=3). (**J**) Microglia translocation between consecutive time points (n=3). On average, the majority of microglia remained within ~10 µm of their original location (white bars). The proportion of microglia that moved within their domain (10–20 µm; orange bars, circles), (20–30 µm; blue bars, squares), or translocated a further distance (>30 µm; gray bars, filled black circles) stayed relatively constant with increasing 4 hr intervals (n=3). (**K**) In one animal, we categorized microglia over time and determined that when normalized by the total number of microglia at the end of the imaging session, the majority of microglia exhibited a doublet morphology at some point during imaging, and very few microglia exhibited either a persistent normal or doublet morphology (did not divide). Scale bar, 50 µm. *Figure 4—source data 1*: Source data for microglia repopulation in mice imaged every 4 hr for 24 hr.

The online version of this article includes the following source data for figure 4:

**Source data 1.** Source data for microglia repopulation in mice imaged every 4 hr for 24 hr.

(*Figure 4*). The simulation was run 500,000 times. A schematic of the model is presented in *Figure 5—figure supplement 1*. In summary, we randomly sampled the number of cells in the population that are eligible for division from a binomial distribution with probability of success equal to the proportion of cells eligible for division and set that equal to the initial day. We counted the number of cells after 24 hr that divided, as well as all the cells in the process of dividing to account for the total number of cells in the final subtotal. In 4 hr intervals (to replicate the 24 hr imaging

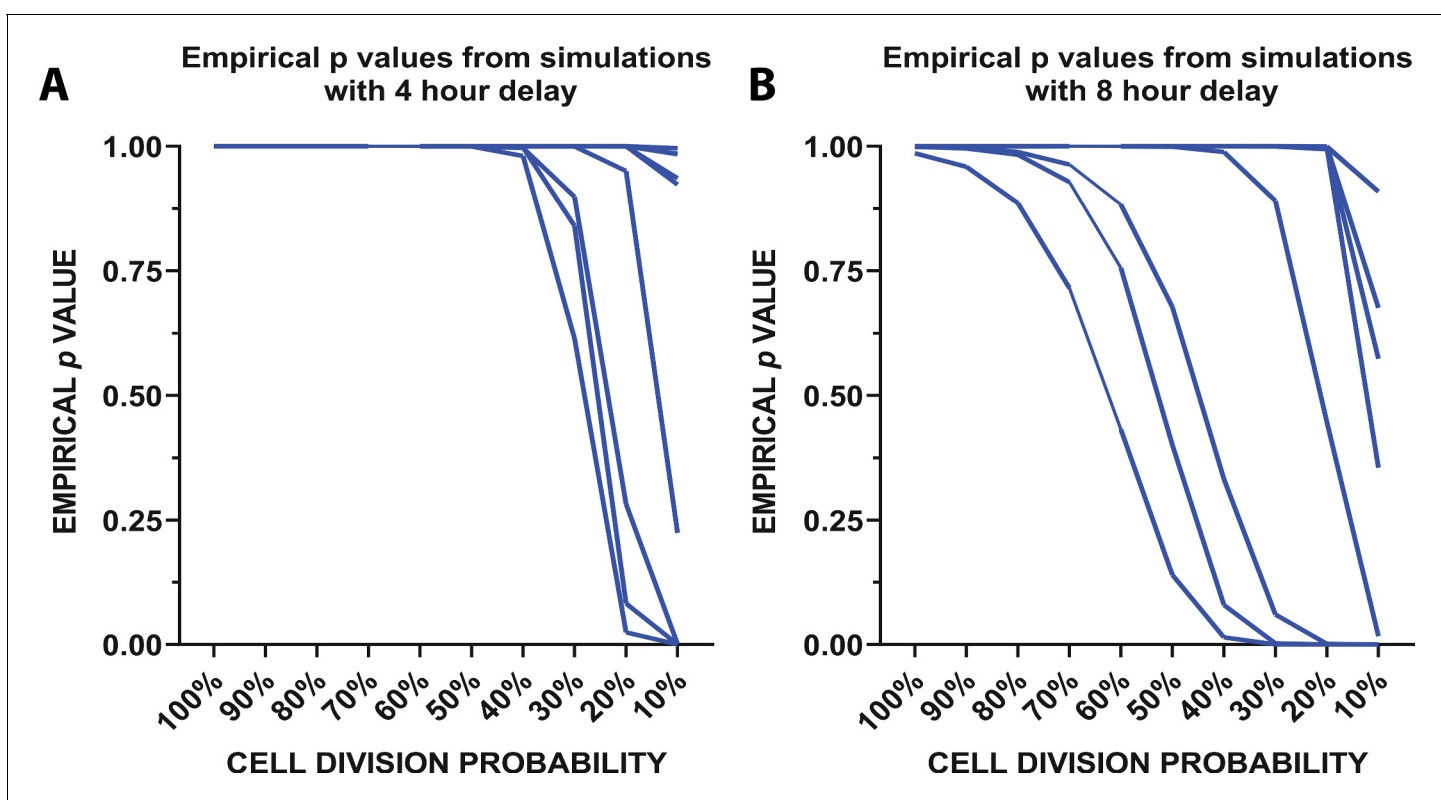

**Figure 5.** Mathematical modeling of microglial repopulation. (**A–C**) Mathematical modeling of microglial repopulation using measured kinetic parameters. Empirical p values from the simulation are plotted for a minimum 4 hr delay (**A**) and 8 hr delay (**B**) between subsequent divisions; each animal is represented by a line (n=10). *Figure 5—source data 1*: Source data for the mathematical modeling of microglial repopulation.

The online version of this article includes the following source data, source code and figure supplement(s) for figure 5:

**Source code 1.** Code to generate the mathematical model simulation.
**Source data 1.** Source data for the mathematical modeling of microglial repopulation.
**Figure supplement 1.** Mathematical model of adult microglia repopulation.

experimental time points), we resampled cells from the population of those eligible for division (total number of cells in circulation minus cells previously selected for division) and corresponding division rates before concluding after 24 hr passed. The empirical p-value is calculated from the simulated data by taking the percentage of the repopulation simulation results where the final number of microglia was greater than or equal to the number of microglia observed on day 3 of the daily imaging paradigm for each animal (*Figure 2*). The empirical p values vs. the cell division probability were plotted from the simulation with a 4 (*Figure 5A*) and 8 hr delay (*Figure 5B*) between divisions. With 50% of cells available for division, it is reasonable to believe that the repopulation we observed from days 2 to 3 in our daily imaging experiments in 11 animals (*Figure 2*) came purely from local doublet cell division if 50% of cells are capable of division (*Figure 5A–B*). In fact, in half of the animals imaged, only 10% of cells would need to be capable of division to repopulate the visual cortex. This data suggests that the local division of remaining microglia is likely responsible for rapid repopulation with newly born microglia capable of further division and without the need for the migration of newly generated microglia from a different site of generation.

## Newly born microglia acquire a hyperramified morphology in the later stages of repopulation

We observed that doublet cells were ramified during division, resulting in new cells that also had ramified morphologies soon after they separated. Because microglial morphology is difficult to quantify in the wide-field imaging in awake animals, we imaged a subset of microglia during repopulation under anesthesia in a separate cohort of animals at high digital zoom to closely quantify the subtle changes in repopulated microglial morphology (*Figure 6A*). We then traced individual microglia and used Sholl analysis to assay the complexity of the arbors (*Figure 6B* and *Figure 6—figure supplement 1*). At the peak of depletion (7 days of PLX), the remaining microglia exhibited a ramified morphology with extensive processes similar to the basal microglia process ramification under steady-state conditions (*Figure 6B–C*) and an increase in soma size that was not statistically significant (*Figure 6F*). In the early phase of repopulation (day 1 – day 3), the newly born microglia appear ramified with secondary processes (*Figure 6B–E*) and have an enlarged soma (*Figure 6F*). After this, a more complex arbor begins to form (*Figure 6B–E*), whereby there is an overshoot in the maximum number of intersections and the integrated area under the Sholl curve at 5 days of repopulation, which coincides with our increase in the number of microglia in our original repopulation data (*Figures 2B* and *5C*). This slight hyperramification is maintained for as long as we imaged (30 days of repopulation; *Figure 6C–D*), despite a return to baseline numbers of microglia (*Figure 3B*) and a return to a smaller soma size and full width half max quantification (*Figure 6E–F*). P2Y12-KO animals were hyporamified as compared to CX3CR1-GFP mice at baseline (*Figure 6—figure supplement 1*) and did not show the hyperramification following repopulation as observed in the CX3CR1-GFP mice (*Figure 6G–K*; *Figure 6—figure supplement 1*), although similar changes in soma size were observed (*Figure 6K*). Therefore, newly born microglia rapidly acquire complex morphologies in a process that is independent of P2Y12. However, newly born microglia exhibit a chronic hyperramification that distinguish them from microglia in control conditions. This hyperramification does not occur in the absence of P2Y12.

## Newly born microglia are dynamic and survey the brain

Given the changes in microglial morphology after repopulation (*Figure 6*), we set out to determine if the differences were also reflected in their dynamics (*Figure 7*). Motility measurements were carried out under anesthesia as adrenergic signaling in the awake condition dampens microglial dynamics (*Stowell et al., 2019a*; *Sun et al., 2019*; *Liu, 2019*), allowing us to reliably track microglial process extension and retraction on the order of minutes (*Figure 7A–C*). PLX treatment led to only a modest decrease in motility (*Figure 7D*), as imaged over a 1-hr period, suggesting that the 20–30% of microglia that remain after depletion, not only have full arbors, but are motile and can effectively interact with elements in the parenchyma. This decrease in motility was accompanied by a small decrease in the instability index, which measures the retraction of stable processes (*Figure 7E*) and a small increase in the stability index, which measures the stabilization of newly extended processes (*Figure 7F*), although neither of these changes reached statistical significance. As new microglia were generated after depletion, microglial motility recovered such that the increase in movement

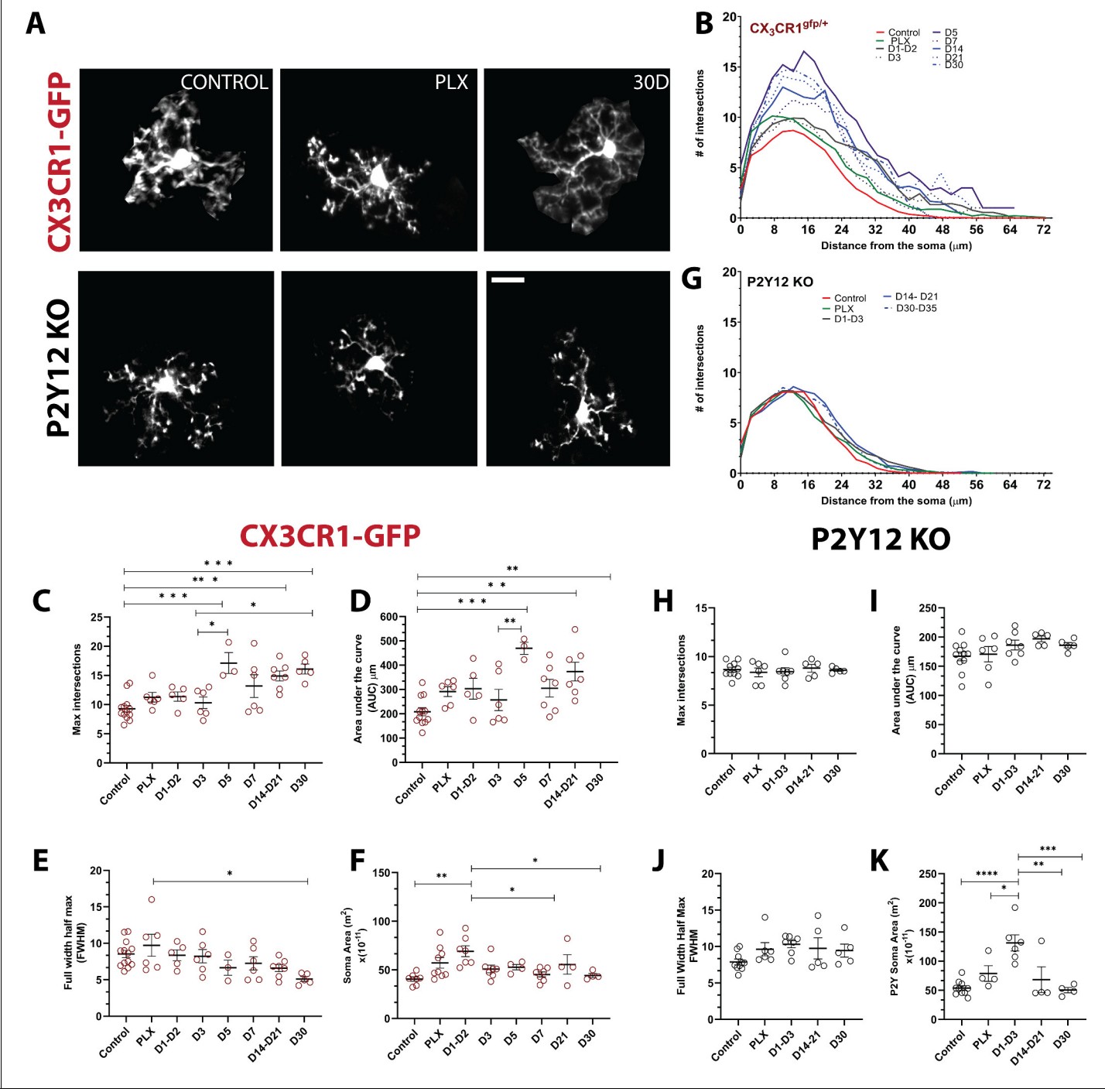

**Figure 6.** Newly born microglia remain in a hyperramified state after repopulation. (**A**) Individual microglia imaged in CX3CR1-GFP and P2Y12-KO mice during control, depletion (7 days of PLX), and repopulation (7 and 30 days). (**B**) Sholl profiles summarizing the morphology of microglia during depletion and repopulation. (**C**) Microglia have an increased number of maximum intersections in the later time points of repopulation; note the increased arborization after day 5 (**d**) Microglia have a greater AUC (Area under the curve) in the later stages of repopulation. (**E**) The full width half max measurement showed a change between the PLX and the later stages of repopulation (D30) n=3–13 animals per group, 2–4 microglia per mouse, (*p<0.05, **p<0.01, ***p<0.001, One-way ANOVA, Dunnett post-hoc test). (**F**) Microglia demonstrated a greater soma area during the early stages for repopulation n=4–9 animals per group, n=3–6 microglia per mouse, (*p<0.05, **p<0.01, One-way ANOVA, Dunnett post-hoc test). (**G**) Sholl profiles for P2Y12-KO mice during depletion and repopulation. There was no significant difference in the maximum intersections (**H**), area under the curve (**I**) and Full width half max (**J**) of the Sholl curves in the absence of P2Y12 (n=4–11 microglia per mouse, ns, One-way ANOVA, Dunnett post-hoc test). (**K**) Microglia demonstrated a greater soma area during the early stages for repopulation n=4–9 animals per group, n=3–6 microglia per mouse, (*p<0.05,

*Figure 6 continued on next page*

*Figure 6 continued*

**p<0.01, ***p<0.001, ****p<0.0001, One-way ANOVA, Dunnett post-hoc test). Graphs show mean ± s.e.m. Points represent individual animals. Red circles (**C–F**) represent CX3CR1-GFP animals while black circles (**H–K**) represent P2Y12-KO animals. Scale bar, 20 μm. *Figure 6—source data 1*: Source data for the morphology and soma area of Newly-born microglia.

The online version of this article includes the following source data and figure supplement(s) for figure 6:

**Source data 1.** Source data for the morphology and soma area of newly born microglia.
**Figure supplement 1.** Comparison of microglial morphology over the course of repopulation in the presence and absence of P2Y12.

correlated with the increase in the number of microglia over time. In addition, during the later stages of repopulation (day 21 day 30), we found no significant difference in basal microglia process motility and long-term stability as compared to baseline (*Figure 7D–F*). Combined with our previous data on microglia repopulation, these data suggest that repopulation can be divided in two stages: early and late repopulation. As microglia mature in the later stages of repopulation (day 7 – day 30), they exhibit more complex arbors suggesting a dysregulation in their morphology. This long-term change in morphology, however, does not impact microglial motility which resembles basal microglial states (*Figure 7D*). To complement the morphology assessment (*Figure 6*), we set out to determine the dynamics of newly born microglia in P2Y12-KO mice. Overall, during the early and late stages of repopulation, we did not observe any changes in motility indices in these mice (*Figure 7G,H,I* and *Figure 7—figure supplements 1–3*). However, P2Y12 KO mice appeared to have lower motility in the early stages of repopulation (D1-D3), as well lower instability and stability indices than CX3CR1-GFP mice (*Figure 7—figure supplements 1–3*).

Finally, to test whether changes in morphology and dynamics of new microglia affect microglial surveillance, we assessed microglial process coverage over one hour (*Figure 7J*). Following 7 days of PLX, the remaining microglia population survey a much smaller area than observed under basal conditions due to their reduced numbers. Microglial surveillance recovers at day 3 of repopulation when microglial numbers return to baseline values and is maintained until day 30 (*Figure 7K*). When surveillance was normalized to the process coverage in the first time point (*t=0mins*) to account for the loss of microglia with depletion, we found that individual microglia surveilled their territory in the same manner throughout the control, depletion, and repopulation conditions (*Figure 7L*). The surveillance pattern was similar in the absence of P2Y12, although P2Y12-KO microglia appeared to be less efficient at surveilling the brain than CX3CR1-GFP microglia (*Figure 7—figure supplement 4*). This difference is largely due to the less ramified arbor in P2Y12 KO microglia, as it was less apparent when surveillance was normalized to the first time point to account for morphology (*Figure 7—figure supplement 5*). Interestingly, it appeared that P2Y12 KO microglia were more efficient at surveying the brain during PLX treatment (*Figure 7M–N*; *Figure 7—figure supplement 5*). Based on these findings, we conclude that newly-born microglia mature within days of repopulation, acquire complex arbors, and survey the environment effectively. Also, while P2Y12 may mediate the microglial landscape under basal conditions (*Eyo et al., 2018*), it likely plays a minor role in newly born microglia ontogeny and maturation (*Figures 2*, *6* and *7*).

## Newly born microglia respond robustly to acute focal tissue injury

To determine whether newly-born microglia can carry out their normal pathological functions, we generated focal laser ablation injuries in the visual cortex using the two-photon laser microscope and quantified the movement of microglial processes toward the site of injury over the course of 1 hr using two separate methods (*Figure 8* and *Video 3*). First, we developed a custom optic flow-based algorithm to calculate the directional velocity of microglia processes moving toward the injury site (*Figure 8E–F*). Analysis of the average of all vectors moving towards the core showed that microglia at all stages of depletion and repopulation responded robustly to laser ablation injury (*Figure 8F,G*). Both the maximum magnitude and the integrated response (area under the curve) were similar across conditions (*Figure 8G,H*). It is interesting to note that, under PLX treatment, the remaining microglia, although less motile, sparse, and randomly organized in the cortex, responded robustly to laser ablation injury with only a slight trend toward a decreased response (*Figure 8F,G, H*). A similar pattern was observed with convergence analysis, which calculates the number of pixels representing microglia processes entering a small ring around the core of the injury site (*Figure 8I*).

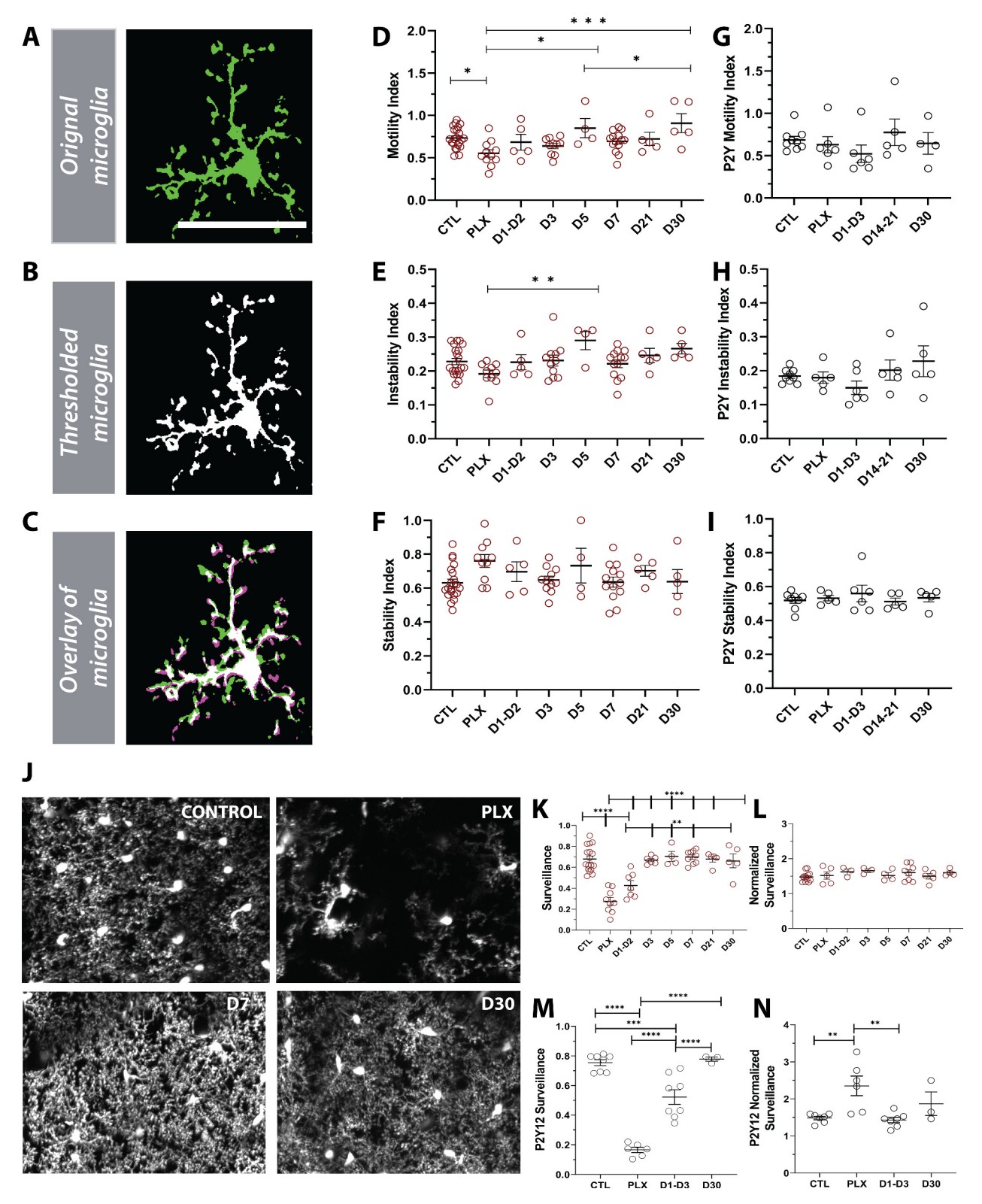

**Figure 7.** Newly born microglia are dynamic and survey the brain. (A) Example of motility analysis showing the original microglia image, (B) Thresholded microglia (C) and a visual representation of microglial motility whereby thresholded imaged from two time points are combined with magenta representing retraction, green representing extension and white representing stable pixels. (D) Quantification of the motility index which compared the gain and loss of pixels across 5 min intervals. Microglia motility was decreased during depletion but recovered quickly with repopulation.

*Figure 7 continued on next page*

*Figure 7 continued*

(E) A similar trend toward decreases during depletion was observed in the instability index which was calculated as the proportion of stable pixels that retracted over the total stable pixels in the first time point. (F) No change in the stability index (the proportion of extended pixels that became stable divided by the total extended pixels in the first overlay) was observed (n=4–17 animals per group, *p<0.05, **p<0.01, ***p<0.001, One-way ANOVA, Dunnett post-hoc test). (G–I) There was no significant change in motility (G), instability (H) and stability (I) indices in the absence of P2Y12 (n=4–9 animals per group, ns, One-way ANOVA, Dunnett post-hoc test). (J) Maximum projection of microglial processes over the hour imaging session to observe surveillance (control, 7 days of PLX, 7 days of repopulation and 30 days of repopulation). (K) Microglial surveillance of the parenchyma dropped as microglia were depleted from the brain but recovered quickly during repopulation as soon as microglial numbers reached control levels (1 day −3 day). (L) Graph of surveillance normalized to the extent of microglial coverage in the first time point (n=4–17 animals per group, **p<0.01, ***p<0.001, ****p<0.0001 One-way ANOVA, Dunnett post-hoc test). (M) Microglial surveillance of the parenchyma dropped as microglia were depleted from the brain in the absence of P2Y12 but recovered quickly as shown in the CX3CR1 controls in (L). (N) Graph of surveillance normalized to the extent of microglial coverage in the first time point in the absence of P2Y12 (n=3–6 animals per group, **p<0.01, ***p<0.001, ****p<0.0001, One-way ANOVA, Dunnett post-hoc test). Graphs show mean ± s.e.m. Points represent individual animals. Scale bar, 50 μm. *Figure 7—source data 1*: Source data for Newly born microglia motility, surveillance during depletion and repopulation.

The online version of this article includes the following source data and figure supplement(s) for figure 7:

**Source data 1.** Source data for newly born microglia motility, surveillance during depletion and repopulation.
**Figure supplement 1.** Comparison of motility over the course of repopulation in the presence and absence of P2Y12.
**Figure supplement 2.** Comparison of changes in the instability index over the course of repopulation in the presence and absence of P2Y12.
**Figure supplement 3.** Comparison of the stability index over the course of repopulation in the presence and absence of P2Y12.
**Figure supplement 4.** Comparison of Surveillance in the presence and absence of P2Y12.
**Figure supplement 5.** Comparison of normalized surveillance in the presence and absence of P2Y12.

Similar to the directional velocity measure, there was no difference in the maximum convergence at 60 min and the area under the curve value at each time point (*Figure 8J,K*). This suggests that the newly-born microglia during the early phases of repopulation can respond robustly to focal laser ablation injury. This is an indication that microglia retain the necessary sensors such as P2Y12 and internal machinery to respond to injury.

## Discussion

In this study, we characterized in detail the dynamics of adult microglia ontogeny and maturation in the visual cortex of mice. Because microglia behave differently in awake vs. anesthetized states (*Stowell et al., 2019a*; *Sun et al., 2019*; *Liu, 2019*), we used an awake in vivo chronic imaging setup to track microglial repopulation. First, we showed that microglia maintain their numbers and territories under basal conditions. Next, we found that microglia can divide rapidly and continuously to repopulate the brain within a few days after partial depletion with the colony stimulating factor one receptor (CSF1R) inhibitor, PLX, in a manner that is largely independent of signaling through the P2Y12R. A mathematical model based on the imaged kinetics of division confirmed that the repopulation was driven by very rapid, local self-renewal by surviving microglia, which transition to a doublet morphology, divide, and migrate apart before undergoing the next cycle of division. Finally, we found that newly born microglia mature rapidly both structurally and functionally and that P2Y12 modulates morphological changes following repopulation.

### The dynamics of microglial division are heterogeneous

Microglia are thought to be long-lived cells, self-renewing slowly and stochastically under unperturbed physiological conditions in the brain with an average lifetime of four years in the human cortex (*Réu et al., 2017*), and 15 months in the mouse (*Füger et al., 2017*). Despite their stability, microglia can rapidly repopulate the brain when their niche has been depleted (*Bruttger et al., 2015*; *Elmore et al., 2015*). While this remarkable capacity for self-renewal has been the subject of intense interest, many questions remain unanswered, regarding the kinetics of microglia self-renewal. By taking a chronic *in vivo* approach coupled with mathematical modeling, we were able to identify and track novel behaviors of residual and dividing microglia *in vivo*. Based on the average cell numbers and the various cell division rates, our simulations of microglia repopulation support the conclusion that the residual cells that remain after depletion can rapidly divide to repopulate the visual cortex.

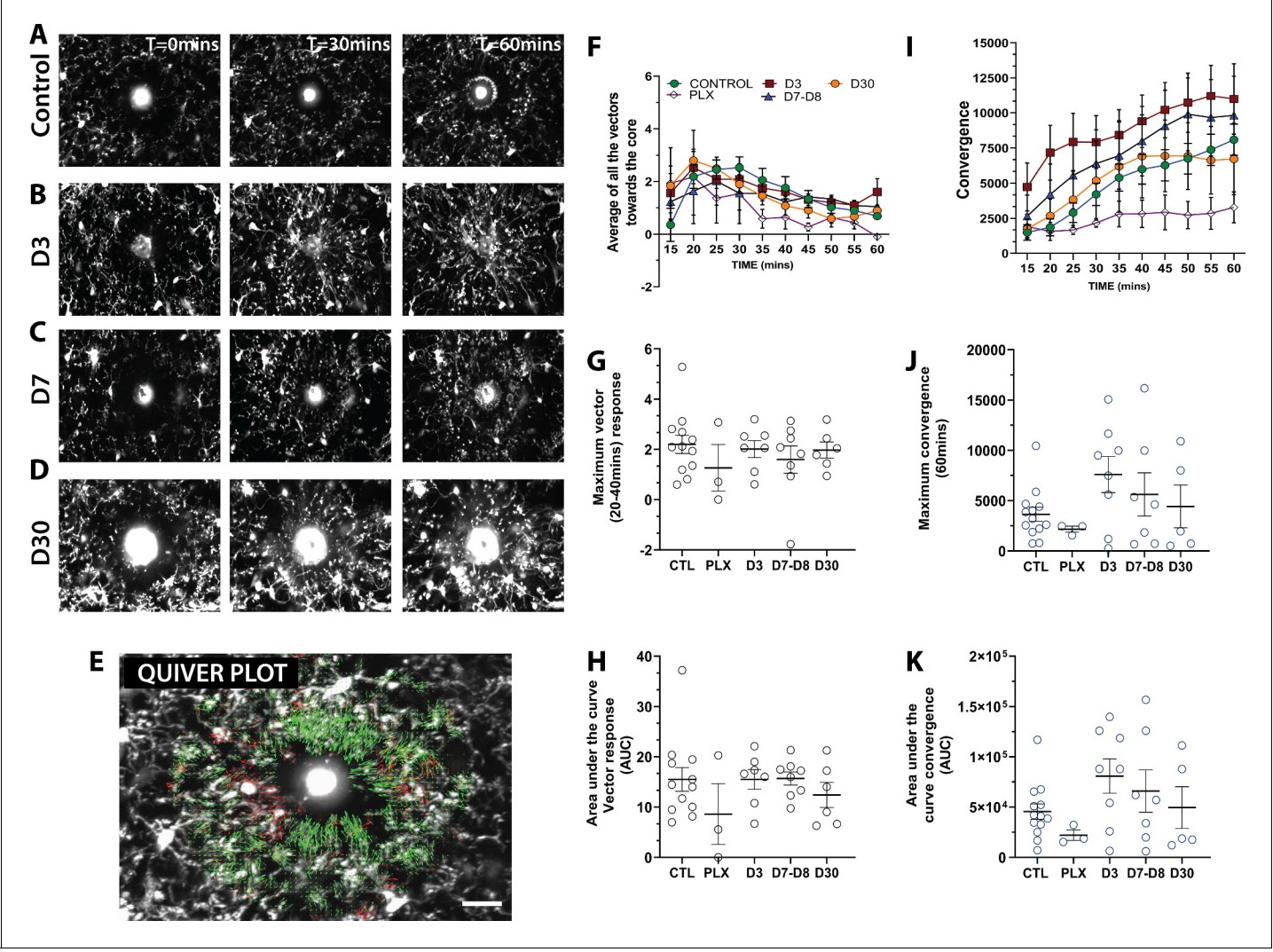

**Figure 8.** Newly born microglia respond robustly to acute laser ablation injury. Example microglial response to focal laser ablation at t=0 min, t=30min and t=55 min after ablation for control (A), 3 days of repopulation (B), 7 days of repopulation (C), and 30 days of repopulation (D). (E) Quiver plot of microglial response. Green arrows correspond to vectors moving towards the core and red arrows correspond to vectors moving away from the core. (F) Graph showing the average vectors moving towards the core following focal laser ablation injury. (G-H; black circles – vector analysis) There were no statistically significant differences observed in the maximum directional response, or the integrated area under the curve over time. The dynamics of the convergence of microglial processes on the injury core n=12 (control), n=3 (PLX), n=8 (7–8 days), n=6 (30 days). (I) Graph showing the convergence towards to core following focal laser ablation injury. (J-K) There was no significant difference in the maximum convergence at 60 min, or the convergence response when assayed using the area under curve of the convergence graphs over time (blue circles – convergence analysis). n=3–12 animals per group, One-way ANOVA, Dunnett post hoc test. Graphs show mean ± s.e.m; ns. Points represent individual animals. Scale bar, 20 μm.

*Figure 8—source data 1*: Source data for newly-born microglia response to acute laser ablation injury.

The online version of this article includes the following source data for figure 8:

**Source data 1.** Source data for newly-born microglia response to acute laser ablation injury.

We observed that during the height of repopulation, ~50% of cells adopted a doublet morphology indicative of an impending division, suggesting that most residual cells are capable of proliferating, and 95% of those cells divided in a 24-hr period. While we also observed these doublet morphologies after repopulation was complete, the division rates at that time were slow with division occurring over a period of days, as seen in the non-depleted brain (*Füger et al., 2017*), and inconsistent with rapid repopulation. In contrast, at the height of repopulation, division could occur very rapidly, with a cell adopting a doublet morphology and undergoing division within 4 hr. These rapidly dividing cells can undergo a secondary division within our 24-imaging period. This secondary

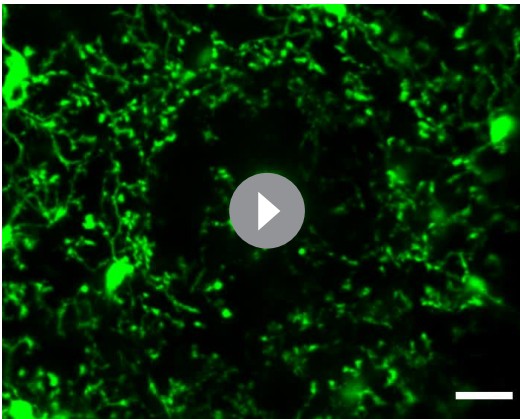

**Video 3.** Example in vivo laser ablation video during depletion and repopulation. Imaging was done in CX3CR1-GFP awake with a 1.5x zoom. Here, we compared a laser ablation during control condition, depletion (PLX), and repopulation (D7) and D30. Microglia responded rapidly during these two conditions. 48 slices total, 12 slices per group. Scale bar, 20 μm.
https://elifesciences.org/articles/61173#video3

division could occur in both cells generated from the first division, suggesting a remarkable capacity for proliferation in residual microglia.

However, the rates of microglia division were not uniform. While some cells divided rapidly and repeatedly, others divided slowly within 8 or 12 hr or remained in the doublet state for the 24-hr period of imaging. While we cannot discern whether these differences in division rates are tied to specific subpopulations of microglia, it is well known that microglia are morphologically and transcriptionally diverse, with high-throughput single-cell transcriptomics identifying distinct subpopulations of microglia with unique molecular signatures that change with age and in response to insults (*Hammond et al., 2019*; *Ayata et al., 2018*; *Li et al., 2019*). This heterogeneity in microglia is also seen across various brain regions (*Hammond et al., 2019*; *Ayata et al., 2018*; *O'Koren et al., 2019*; *De Biase et al., 2017*). Similar heterogeneity in the residual microglial population may play a role in the dynamics of repopulation, contributing to a heterogeneity in the capacity to divide. For instance, it has been suggested that the remaining microglia after depletion are a specialized CSF1R-inhibitor resistant population of Galectin-3-progenitor-like microglia (*Zhan et al., 2020*). We did not see a correspondence between our doublet morphologies and Galectin-3 expression, which supports the idea that proliferating cells can be Galectin-3 positive or negative (*Zhan et al., 2020*), although other factors could define proliferating microglial populations. A similar scenario has been observed in the retina where microglia repopulate from a distinct source and migrate through the optic nerve to repopulate the rest of the retina (*Huang et al., 2018a*). While more detailed studies will be needed to determine the extent of microglial heterogeneity after PLX treatment, characterizing the heterogeneity of repopulating microglia may provide new avenues for understanding the phenotypes of adult-generated vs. yolk sac born microglia.

## Newly generated microglia rapidly recapitulate functional features of endogenous microglia

We show that newly born microglia acquire mature characteristics such as baseline motility, surveillance, and response to acute laser ablation remarkably quickly. In fact, dividing doublet microglia have an extensive, motile arbor which is maintained through the division such that the two resulting cells already have extensive processes which surveil the environment and can respond to injury. While microglial expression in newly repopulated microglia recapitulates some aspects of neonatal microglia expression patterns (*Zhan et al., 2019*; *Zhan et al., 2020*), we did not observe the typical morphologies of microglia or changes in process dynamics and responses to injury that could indicate a less mature phenotype. Developing microglia transition from an ameboid morphology to a ramified morphology at around post-natal day 30 (*Orłowski et al., 2003*), and while microglial dynamics have not been extensively studied in development, early life microglia have few processes and are migratory (*Kierdorf et al., 2013*; *Hoeffel et al., 2015*; *Ginhoux et al., 2013*). Since newly born microglia did not exhibit ameboid morphologies, and somas tended to not move more than 10–20 μm, this suggests that if microglia do revert to a developmental program when they are first generated in the adult brain, this program does not significantly alter their physiological functions. Alternately, they may go through an accelerated maturation program, as expression also matures much faster than the stepwise development of microglia over a period of weeks in early life (*Zhan et al., 2019*; *Bennett et al., 2016*; *Matcovitch-Natan et al., 2016*).

## Changes in microglial morphology in newly born microglia

While many studies have shown that after 30 days of repopulation, new microglia largely adopt the gene expression profile of control microglia (*Elmore et al., 2015*; *Zhan et al., 2019*), several observations in our study suggest that adult born microglia may not be identical to their yolk-sac-generated counterparts. We found that after repopulation, microglia morphology changed with a hyper-ramification of the processes and that more microglia adopted a doublet-like morphology with an elongated cell body. As these changes persisted for a month after the cessation of PLX, it may be that newly born microglia take on a unique phenotype that persists long after microglia are mature. While hyper-ramification of the microglial arbor was observed in CX3CR1-GFP animals, it was not apparent in P2Y12-KO animals, possibly due to the general effects of P2Y12 loss on ramification at baseline (*Sipe et al., 2016*). Hyper-ramification of microglia is typically associated with periods of enhanced plasticity such as that following monocular deprivation in the developing visual cortex (*Huang et al., 2018b*) and with chronically stressed animals in a model of posttraumatic stress disorder (PTSD) (*Smith et al., 2019*) where microglia transition from their normal ramified morphology to a hyper-ramified state, characterized by an increase in primary and secondary processes. A similar microglial response, which has been referred to as arrested hyper-ramification, is also observed during aging and in response to specific neurotoxins (*Streit et al., 1999*; *Streit et al., 2004*). Whether this chronic morphological change is a product of the rapid repopulation and efforts to restore homeostasis, a direct result of PLX treatment, or a general phenotype of newly-born microglia is unclear.

The presence of a persistently increased number of microglia with elongated, doublet-like cell bodies, after repopulation, could be indicative of further morphological alteration of newly born microglia, but it could also suggest an increased capacity for division as similar doublets have been identified as a source of dividing cells in another study (*Füger et al., 2017*). If Galectin-3+ microglia that remain after depletion *Zhan et al., 2020* have increased capacity for division, they may divide to produce new microglia that are distinct from yolk sac-generated microglia and rapidly take on mature characteristics, contributing to a different phenotype of adult vs. yolk-sac born microglia. The Galectin-3+ cells may remain in the population as persistent doublet cells primed for division and self-renewal. Thus, while adult born microglia appear to rapidly take on their mature roles in the brain, their characteristics may differ in subtle ways from microglia that are generated in the yolk sac and populate the brain in development.

## Newly generated microglia chronological versus biological age

Functional characterization of newly born microglia may provide valuable clues for determining treatments for neurodegenerative and neurodevelopmental diseases where microglia are often thought to be dysregulated (*Perry et al., 2010*; *Keren-Shaul et al., 2017*; *Krasemann et al., 2017*). It is possible that efficient rejuvenation of old 'senescent' microglia in the diseased brain with new microglia through depletion and repopulation can generate a new population of microglia with improved functions (*Rice et al., 2015*; *Spangenberg et al., 2016*; *Elmore et al., 2018*). However, it is important to recognize that it remains unclear whether the chronologically younger newly born cell is also biologically younger. In fact, we do not yet understand the biological process of microglia repopulation, and it is not known whether the division of microglia is symmetrical or asymmetrical. To determine whether a newly divided cell is in fact, 'rejuvenated,' the 'maturity' of that cell will need to be tested directly. This could be assessed by looking at transcriptional and epigenetic signatures and comparing to similar studies done in development (*Hammond et al., 2019*; *Matcovitch-Natan et al., 2016*). Newly born microglia may go through a stepwise expression program that partly recapitulates development, giving them younger functional characteristics, or they may acquire all the epigenetic and cytoplasmic features of their 'aged' parent cell, leading to repopulation but not 'rejuvenation' of the microglia niche (*Zhan et al., 2019*). This latter option could also explain why the newly-born cells acquire mature characteristics and functions early in the repopulation phase.

## Intrinsic and extrinsic mechanisms of microglia repopulation

A combination of exogenous and endogenous signaling mechanisms may contribute to microglia rapid repopulation and spatial organization. Recent evidence has shed light onto some of the extracellular signals that shape repopulation, including those that regulate microglia proliferation such as

colony stimulating factor 1 (CSF-1) and interleukin one receptor (IL1R), and those that regulate migration such as fractalkine through its receptor, CX3CR1 (10, 11, 20). A gradient or threshold of these factors may start intracellular cascades in microglia that trigger division and migration, such as NF-KB signaling, which has been shown to be important for microglia to repopulate fully (*Zhan et al., 2019*). In addition, we show that the P2Y12 receptor does not modulate newly born microglial maturation (*Figures 6* and *7F*) despite effects of P2Y12 loss on microglia ramification (*Figure 6—figure supplement 1*; *Haynes et al., 2006*; *Sipe et al., 2016*), and it does not affect doublet formation, repopulation, or the acquisition of microglial territories (as reflected by changes in nearest neighbor quantification) in the visual cortex (*Figures 2* and *3*). This is surprising given that P2Y12 modulates microglial translocation in baseline conditions (*Eyo et al., 2018*). All together, this suggest that microglia repopulate and mature in a P2Y12-independent manner. The progressive distribution of microglia to achieve tiling during repopulation is most likely maintained by lateral inhibition mechanisms and homeostatic microglia-specific genes such as Sal1 and Mafb—which both maintain microglia in a ramified nonclustered state (*Matcovitch-Natan et al., 2016*; *Buttgereit et al., 2016*). Because microglial numbers overshoot their target during the early stages of repopulation, it is also likely that astrocytes may be recruited to phagocytose excess newly-born microglia, and these cells may contribute to the extracellular environment that promotes microglial migration and maturation. Altogether, an interplay of diverse factors likely contributes to microglia homeostasis following depletion, which may include endogenous mechanisms at play in specific populations of microglia, including Galectin-3+ cells, as well as exogenous factors that come from microglia, neurons, and astrocytes and work together to re-establish microglia numbers in the adult brain.

## Concluding remarks

Together our results build on current adult microglia ontogeny research showing that rejuvenation of adult microglia is driven by stochastic doublet cell division locally during repopulation, with newly born cells rapidly able to take on their roles in the brain. Loss of P2Y12 impacts microglial morphologies during repopulation but seems to play a minor role in microglial division. In addition, our mathematical modeling supports the idea that division of residual microglia alone can fill the microglia niche following depletion. While our in vivo tracking approach illuminates the rapid and heterogeneous dynamics of microglial repopulation, future studies will be needed to fully understand the mechanisms that lead to the generation of new microglia and their ability to adopt their mature features. Such studies will provide a deeper understanding of the important role microglia play in homeostasis and disease.

# Materials and methods

**Key resources table**

| Reagent type (species) or resource | Designation | Source or reference | Identifiers | Additional information |
|---|---|---|---|---|
| Strain, strain background | CX3CR1-GFP, C57/Bl6 background (Mouse) | JAX | RRID:IMSR_JAX:005582 Jax stock #005582 | B6.129P2(Cg)-*Cx3cr1*^*tm1Litt*/J Male and Females used in our study |
| Strain, strain background | P2Y12 KO, C57/Bl6 background (Mouse) | Courtesy of Maiken Nedergaard lab via a Materials transfer agreement | | Mice were generated by the Conley lab doi:10.1172/JCI17864. Male and Females used in our study |
| Chemical compound, drug | PLX5622 drug | Research Diets, New Jersey, USA | AIN76A-D1001i | PLX5622 drug from Plexxikon Inc, Berkeley, CA, USA |
| Chemical compound, drug | Fentanyl (0.05 mg/kg) | University of Rochester Medical Center Pharmacy | | |
| Chemical compound, drug | Midazolam (5.0 mg/kg) | University of Rochester Medical Center Pharmacy | | |

*Continued on next page*

*Continued*

| Reagent type (species) or resource | Designation | Source or reference | Identifiers | Additional information |
|---|---|---|---|---|
| Chemical compound, drug | Dexmedetomidine (0.5 mg/kg) | University of Rochester Medical Center Pharmacy | | |
| Other | 3 mm Biopsy punch | Integra, Plainsboro, NJ, USA | Cat#: 33–32 | |
| Other | 0.5 mm drill bit | FST, Foster City, CA | Cat#: 19007–05 | |
| Other | UV Glue | Norland Optical Adhesive, Norland Inc, Cranbury, NJ, USA | Cat#: NOA 61 | |
| Other | Loctite 404 Glue | Henkel Corp, Bridgewater, NJ, USA | Cat#: A-A-309 | |
| Other | Headplate | Emachine shop | | Design courtesy of Mriganka Sur |
| Other | C and B Metabond Dental Cement | Parkell Inc Brentwood, NY, USA | SKU #: S380 | |
| Other | Ti: Sapphire | Mai-Tai Spectraphysics Olympics, Center Valley, PA | | Custom built two photon microscope |
| Antibody | Anti-Iba-1 (Rabbit polyclonal) | Wako | RRID:AB_839504 Cat # 019–19741 | IF(1:1000) |
| Antibody | Anti-GFAP Glial Fibrillary Acidic Protein (Mouse monoclonal) | Sigma | RRID:AB_477010 Cat# G3893 | IF(1:500) |
| Antibody | Anti-Galectin3 (Goat polcylonal) | R and D systems | RRID:AB_2234687 Cat# AF1197 | IF(1:1500) |
| Antibody | AlexaFlour 488 (Donkey Anti-Rabbit) | Invitrogen | RRID:AB_2762833 Cat# A32790 | IF(1:500) |
| Antibody | AlexaFlour 594 (Donkey Anti-Mouse) | Invitrogen | RRID:AB_2762826 Cat# A32744 | IF(1:500) |
| Antibody | AlexaFlour 594 (Donkey Anti-Goat) | Invitrogen | RRID:AB_2762828 Cat# A32758 | IF(1:1000) |
| Commercial assay or kit | MILLIPLEX MAP Mouse TH17 Magnetic Bead Panel (Mouse) | Millipore, Sigma | MTH17MAG-47K | |
| MATLAB | MATLAB and Statistics Toolbox Release 2012b, The MathWorks, Inc, Natick, | Massachusetts, United States | RRID:SCR_001622 | |
| GraphPad Prism | GraphPad Software, https://www.graphpad.com'. | San Diego, California USA, | RRID:SCR_002798 | |
| R | R Project for *Statistical* Computing | | RRID:SCR_001905 | |

## Experimental animals

All animal works were performed according to the approved guidelines from the University of Rochester, Institutional Animal Care and Use Committee and conformed to the National Institute of Health (NIH). Animals were housed in a 12 hr light/12 hr dark cycle with food ad libitum. Three- to 6-month-old male and female heterozygous (CX3CR1-GFP) (RRID:IMSR_JAX:005582) GFP reporter mice expressing GFP under control of the fractalkine receptor (CX3CR1) promoter (*Jung et al., 2000*) and P2Y12-KO mice were used for in vivo live imaging of visual cortex microglia repopulation and dynamics. All mice were derived from and maintained on a C57/Bl6 background.

## Microglia depletion

Diet (AIN-76A-D1001i, Research Diets, New Jersey, USA) containing 1200 mg/kg PLX5622 (Plexxi-kon Inc, Berkeley, CA, USA) was given to mice as the sole food source for 1–2 weeks to deplete microglia. Control diet with the same base formula but without the compound was given to the control group (Bl6 mice in Supplementary Figure 1 and 2 that are not exposed to PLX) and during the repopulation phase for 3–4 weeks.

## Cranial window surgery

Animals were anesthetized using a fentanyl cocktail (i.p.) during the cranial window implantation surgical procedure. The fentanyl cocktail consisted of fentanyl (0.05 mg kg$^{-1}$), midazolam (5.0 mg kg$^{-1}$), and dexmedetomidine (0.5 mg kg$^{-1}$). Body temperature was maintained at 37°C with a heating pad and aseptic technique was maintained during all surgical procedures. Mice were placed in a stereotaxic frame and head-fixed for cranial window surgeries. Hair was removed and the skull was exposed through a scalp incision. A 3 mm Biopsy punch (Integra, Plainsboro, NJ, USA) was used to create a circular score on the skull over V1. A 0.5 mm drill bit (FST, Foster City, CA) was used to then drill through the skull for the craniotomy, tracing the 3 mm score. A 5 mm coverslip attached to a 3 mm coverslip (Warner Instruments, Harvard Bioscience, Hamden, CT) by UV glue (Norland Optical Adhesive, Norland Inc, Cranbury, NJ, USA) was then slowly lowered down into the craniotomy (3 mm Side down). The coverslip was secured with Loctite 404 glue (Henkel Corp, Bridgewater, NJ, USA). A custom headplate produced by emachine shop (http://www.emachineshop.com) (designs courtesy of the Mriganka Sur Lab, MIT) was kept in place with C and B Metabond Dental cement (Parkell Inc, Brentwood, NY, USA). The dental cement was used to cover exposed skull and keep the headplate in place. Mice were administered slow-release buprenex by veterinary staff (s.q. mg kg-1 every 72 hr) and monitored for 72 hr.

## Two-photon microscopy

A custom two-photon laser-scanning microscope was used for in vivo imaging (Ti: Sapphire, Mai-Tai, Spectraphysics; modified Fluoview confocal scan head, 20x water immersion objective lens, 0.95 numerical aperture, Olympus, Center Valley, PA). Excitation for fluorescent imaging was achieved with 100-fs laser pulses (80MHz) tuned to 920 nm for GFP with a power of ~30–40 mW measured at the sample. Fluorescence was detected using a photomultiplier tube in whole-field detection mode using a 580/180 filter. Images were collected from 20 μm to 300 μm into the brain. For repeated imaging, blood vessels were used as gross landmarks and stable microglia were also used as fine landmarks to re-identify the correct region for imaging. Image analysis was done offline using ImageJ and Matlab with custom algorithms as described in *Stowell et al., 2019a* and available at https://github.com/majewska-lab (*Stowell et al., 2019b*).

## Awake and anesthetized imaging sessions

For awake imaging, mice were trained and habituated to awake imaging during the 2-week recovery from cranial window surgeries. Mice were head fixed under the microscope for 10 min on the first day and the time spent head fixed was increased by 5 min daily. Habituation sessions were terminated if mice exhibited signs of discomfort such as excessive movement and vocalization. Training was complete once mice tolerated 4 consecutive days of head fixation. The imaging session did not exceed 60 min daily. For imaging of microglial morphology and motility, a separate cohort of mice was used and anesthetized using Fentanyl Cocktail. Fentanyl cocktail contained Fentanyl (0.05 mg per kg, intraperitoneally), midazolam (5.0 mg per kg, intraperitoneally), and dexmedetomidine (0.5 m per kg) premixed and given intraperitoneally for each anesthetized imaging session.

## Microglia migration analysis

Two-photon XYZ images of microglia were collected. For analysis, microglia within a volume of 800 X 600 X 100 μm (approximately 60 to 160 μm from the surface of the brain) were used. The images were processed and aligned using Fiji ImageJ. The centroid for each microglia was identified and the x, y, and z coordinates were recorded. Images were aligned over consecutive days using the blood vessels as gross landmarks. A custom algorithm was created to quantify the distances between individual microglia and the remaining microglia in the image. The minimum value was

then used as the 3D nearest neighbor value for that microglia and nearest neighbor values were averaged together for all microglia in a single animal at each time point.

Distances were calculated as shown below:

$$\text{distances}(i,j) = \text{sqrt}(\text{xaxis}(i) - \text{xaxis}(j))^2 + (\text{yaxis}(i) - \text{yaxis}(j))^2 + (\text{zaxis}(i) - \text{zaxis}(j))^2$$

i, j corresponds to number of microglia in the matrix used to calculate the NN.

## Microglial translocation

The location of selected microglia from the first time point were identified, and the XYZ coordinates for the microglia were recorded for the first day. The 3D Nearest neighbor values were calculated as shown above except the location of a microglia was compared to the location of all microglia at another time point. Usually, translocation was measured relative to Day1 of imaging or the time point immediately prior.

## Mathematical modeling of microglial repopulation

We began with three parameters:

1. $N_0$, the number of cells in the population on day 2 of the daily imaging paradigm,
2. $p$, a specified probability that a cell will undergo division,
3. $R$, the observed division rates in experimental data.

Using R (RRID:SCR_001905), the algorithm was then initialized with the following values (*Figure 5—source code 1*). The number of cells chosen to divide from is given by $D_0$ and were selected by sampling from a binomial distribution of size $N_0$ with probability $p$. Each cell selected for the division was assigned a division rate by generating a sample $S_0$ of size $D_0$ from $R$. This process was repeated every 4 hr over a 24-hr period (*Figure 5—figure supplement 1*). We also considered cells to be ineligible for the repeated division for a period of 4 hr after having divided. Using the specified notation and letting $i = 1, 2, 3, 4, 5, 6$ and $t_n = 4n$, we can denote the number of microglia in the population during the $i^{th}$ interval by $N_i$, where

$$N_i = \begin{cases} N_0 - D_0, & \text{if } i=I \\ N_{i-1} - D_{i-1} + 2\sum_{j=0}^{i-2}\sum_{k=1}^{D_j} I(S_{jk} = t_{i-1-j}), & \text{if } i>I. \end{cases}$$

The final number of cells counted in the population was given by the sum of cells that have completed division and those that are still undergoing division, expressed by

$$N_{final} = N_6 + \sum_{j=0}^{5}\sum_{k=1}^{D_j} I(S_{jk} > t_{5-j}).$$

## Microglia morphology

For microglial morphology analysis, 2–4 microglia were selected per animal in an imaging session. For each microglia selected, a z-projection was created using FIJI. All microglia processes were manually traced, thresholded to generate a binarized outline of the process arbor, filtered to remove artifacts, and analyzed with an automated Sholl Analysis plugin (provided by the Anirvan Ghosh laboratory, University of California, San Diego). The maximum number of intersections and the area under the curve (AUC) of the Sholl profile was analyzed to determine microglia arbor complexity.

## Microglia motility and surveillance

For motility analysis, XYZT images consisting of 40-μm-deep z-stacks were collected every 5 min, 12 times for a total of 60 min. Single-image 10 μm Z-projections were created for each time point, and lateral motion artifact was corrected using the StackReg and TurboReg functions (http://bigwww.epfl.ch/thevenaz/stackreg/). After thresholding and binarizing of the maximum intensity projections for all time points together, overlays of consecutive time points (0–5 min, 5–10 min, etc.) were made, so that white pixels represented stability. A custom Matlab algorithm was used to compare pixels across individual time points and across consecutive time points to generate a motility index (defined as the sum of all changed pixels divided by the unchanged pixels). Additional indices were

generated, including stability, as the proportion of extension pixels (green) in one overlay that became stable (white) in the subsequent overlay divided by the total extension (green) pixels in the first overlay. Conversely, an instability index was calculated as the proportion of stable (white) pixels in one overlay that became retracted (magenta) in the subsequent overlay divided by the total stable (white) pixels in the first overlay.

For the surveillance ratio, we z projected all 12-time points and compiled them into a stack. We then aligned the stacks and modulated the brightness/contrast to ensure that all processes were visible, and background was minimal. The z-projected files were then thresholded. The thresholding parameters were chosen to capture most of the processes while minimizing background pixilation. The thresholded time points were used to calculate the area surveyed (surveillance ratio) by microglia during the 1 hr imaging session. This was done by calculating the total number of pixels representing microglia divided by the total pixels in the field of view. This number was further divided by the proportion of pixels representing microglia in the 1st time point to generate the normalized surveillance.

## Microglia soma area quantification

Two to 10 microglia were selected at random from high-magnification images. Single-image 10 μm Z-projections were created for each time point, and lateral motion artifact was corrected using the StackReg and TurboReg functions (http://bigwww.epfl.ch/thevenaz/stackreg/). The polygon tool was used to outline the soma of microglia, and the area was calculated in microns (μm).

## Laser ablation

Laser ablation injuries were created by running a point scan for 8 s at 780 nm using ~75 mW at the sample. The microglia injury response was imaged by collecting z-stacks of 50–90 μm every 5 min. For analysis, Z-projections were all comprised of 10 μm of the stack, encompassing the approximate center of the ablation core. The file was converted to an AVI and subjected to analysis by a custom MATLAB script designed to calculate the movement of microglial processes towards the ablation core. Briefly, for each pixel at each time point, the script generates a vector that estimates the magnitude and direction of motion of the pixel utilizing the Farneback method for estimating optic flow. For analysis, we only included vectors larger than 5 pixels of motion, which were directed toward the ablation core to minimize noise. The magnitude of all the vectors at each time point was summed and normalized to the total number of pixels in the image. For the convergence analysis, the number of pixels that enter the core of the focal injury was summed. We quantified the area under the curve and the maximum value of the normalized magnitude over the 1 hr session.

## Histology

Whole brains were perfused with 0.1M PBS and fixed overnight with paraformaldehyde (4%). The tissue was cut on a freezing microtome (Microm; Global Medical Instrumentation) at 50 μm. For immunohistochemistry, sections were rinsed, and endogenous peroxidase activity and nonspecific binding were blocked with a 10% BSA solution. Sections were then incubated in primary antibody solution to detect microglia (24 hr, 4°C, anti-Iba-1, 1:1,000, Wako 019–19741) followed by a secondary antibody solution (4 hr, RT, AlexaFluor 488, 1:500, Invitrogen), mounted and coverslipped. To determine microglial depletion and repopulation, primary visual cortex sections were imaged on a Zeiss LSM 510 confocal microscope (Carl Zeiss). For each section, a 10 mm z stack in the center of the tissue was collected with a z step of 1 μm at ×40 magnification. The analysis was performed offline in ImageJ. Z stacks were smoothed and compressed into a single z projection. Microglial cell bodies were marked in ImageJ using the paintbrush tool. Results from four to five sections per animal were averaged. Density was calculated as the number of microglia per area in the visual cortex for PLX depleted and control groups.

For the quantification of astrocytes, tissue was processed for histology as described above. Sections were incubated in a primary antibody solution (overnight, 4°C, Anti-Glial Fibrillary Acidic Protein (GFAP), 1:500, Sigma G3893, clone G-A-5) followed by a secondary antibody solution (4 hr, RT, AlexaFluor 594, 1:500, Invitrogen), mounted and coverslipped. To determine astrocyte coverage, somatosensory cortex sections were imaged on a Zeiss LSM 510 confocal microscope (Carl Zeiss). For each section, a 10 μm z stack in the center of the tissue was collected with a z step of 1 μm at

×40 magnification. The analysis was performed offline in ImageJ. The images were thresholded, and the total number of white pixels were then measured. Results from three to four sections per animal were averaged.

Immunohistochemical assessment of repopulated microglia was carried out in brains of CX3CR1-GFP animals received control chow for 2.5 days after 7 day PLX5562 or control that were never exposed to PLX (n=four for each treatment, mixed sex). Tissue was processed for histology in similar manner as described above. Sections were incubated in goat-anti-Galectin 3 (1:1500, R and D Systems AF1197) at 4°C overnight, then washed and incubated in AlexaFluor 594 (1:1000, Invitrogen) secondary at RT, subsequently mounted and coverslipped. Consistent with our in vivo imaging, primary visual cortex sections were imaged (three sections/animal) on a Nikon AR1 HD laser scanning confocal microscope (Nikon). Two or three non-overlapping images were collected per V1; using a 40X water-submersion objective lens and imaged at 0.5 μm z-step using the same laser setting across all sections. Analysis was performed using ImageJ, and statistical analysis was completed in R (R Core Team 2021). For image analysis, the intensity of Galectin-3 was measured in doublets, singlets, and control microglia separately. In addition, microglial soma morphology was also assessed using several parameters: perimeter, area, long axis, and short-axis as well as elongation (long/short axis ratio). Data were processed in R and plotted using Prism (GraphPad).

## Immunoassay

CX3CR1-GFP mice (n=seven for each treatment, mixed-sex) were fed PLX5622 chow or control chow for seven consecutive days. On the 7th day, mice were sacrificed using Euthasol (1:10 dilution) and perfused with 0.15M PBS containing 0.5% sodium nitrite (weight/volume) and 2 I.U. heparin/mL. Cortices were dissected from one hemisphere then immediately frozen in ice-cold isopentane. Frozen cortices were weighed and homogenized in T-PER (pH=7.6, 25 μL/mg, Thermo Fisher) for 30 s, vortexed, and subsequently sonicated for an additional 30 s. Cortical lysates were centrifuged for 5 min at 14,000 g twice; each time, clarified supernatants were saved, and the pellets were discarded. Concentrations of inflammatory modulators in homogenates were measured using MILLIPLEX MAP Mouse TH17 Magnetic Bead Panel (Millipore, Sigma) as per manufacture recommendations. The following cytokines were assessed: IL-1β, IL-4, IL-6, IL-10, TNFα. For all cytokines, a 5-Parameter logistic model was used to fit the standard curves for absorbance values.

## Statistics

Statistical comparisons were made between animal and treatment groups using Prism eight software (GraphPad, San Diego, CA). Samples sizes are similar to those reported in the field and were based on previous experiments. All *n* represents individual animals. Exclusion criteria were discussed prior to analysis and included (1) poor image quality (either due to excessive movement artifact or low signal to noise) as determined qualitatively prior to quantification; (2) the health of the animal (defined as hunched posture and limited movement) as observed during imaging; (3) incomplete imaging sessions. All microglia analyzed in each animal were averaged to generate a single value per animal. Because not all animals could be imaged at all time points, we used a one-way ANOVA and treated each measurement at different time points as independent, which was supported by the examination of residuals. All values reported are the mean ± s.e.m. For all analyses, $\alpha = 0.05$. Two-tailed unpaired or paired *t*-tests and one-way or two-way ANOVA with or without repeated measures (ANOVA) with Tukey post hoc comparisons were used to compare cohorts where appropriate. The exact p values are provided in a supplementary file. The data met the assumptions of normality and equal variances as tested by Prism eight as part of the statistical analyses.

## Acknowledgements

We thank J Olschowka and K O'Banion for sharing PCR resources. We thank the Ghosh laboratory for the Sholl Analysis Fiji plugin. This work is supported by grants from the National Institute of Health (NIH): F99 NS108486-02 (MSM), R01 EY019277, RO1 NS114480 and R21 NS099973 (AKM), and NSF 1557971 (AKM). We thank Plexxikon for providing PLX5622.

# Additional information

## Funding

| Funder | Grant reference number | Author |
|---|---|---|
| NIH Blueprint for Neuroscience Research | F99 NS108486-02 | Monique S Mendes |
| National Institute of Neurological Disorders and Stroke | R01 EY019277 | Ania K Majewska |
| National Institute of Neurological Disorders and Stroke | RO1 NS114480 | Ania K Majewska |
| National Institute of Neurological Disorders and Stroke | R21 NS099973 | Ania K Majewska |
| National Science Foundation | 1557971 | Ania K Majewska |

The funders had no role in study design, data collection and interpretation, or the decision to submit the work for publication.

## Author contributions

Monique S Mendes, Conceptualization, Data curation, Formal analysis, Funding acquisition, Validation, Investigation, Visualization, Methodology, Writing - original draft, Project administration, Writing - review and editing; Linh Le, Data curation, Formal analysis, Methodology, Writing - review and editing; Jason Atlas, Data curation, Formal analysis, Writing - review and editing; Zachary Brehm, Matthew N McCall, Formal analysis, Methodology, Writing - review and editing; Antonio Ladron-de-Guevara, Evelyn Matei, Cassandra Lamantia, Formal analysis; Ania K Majewska, Conceptualization, Resources, Supervision, Funding acquisition, Validation, Writing - original draft, Writing - review and editing

## Author ORCIDs

Monique S Mendes (iD) https://orcid.org/0000-0001-9800-5923
Antonio Ladron-de-Guevara (iD) http://orcid.org/0000-0003-1093-2509
Ania K Majewska (iD) https://orcid.org/0000-0002-2167-6849

## Ethics

Animal Experiments: All animal work was performed according to the approved guidelines from the University of Rochester, Institutional Animal Care and Use Committee and conformed to the National Institute of Health (NIH). Animals were housed in a 12-hour light/12-hour dark cycle with food ad libitum. Mice were housed in cages on a standard 12:12 hour light/dark cycle with food and water ad libitum. All animal experiments were carried out in accordance with the National Institutes of Health Guide for the Care and Use of Laboratory Animals. Ethical approval number: UCAR: 2008-111; expires Dec. 1, 2023.

## Decision letter and Author response

Decision letter https://doi.org/10.7554/eLife.61173.sa1
Author response https://doi.org/10.7554/eLife.61173.sa2

# Additional files

## Supplementary files

• Transparent reporting form

## Data availability

All data generated or analyzed during this study are included in the manuscript and supporting files. All MATLAB codes are available at https://github.com/majewska-lab.

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
