## [Decision Letter]

**Acceptance summary:**

All reviewers highlighted the interest of your study that provides an elegant longitudinal study and mathematical modeling of microglial self-renewal and maturation. Your response and revised manuscript fully answers the reviewers comments on the mathematical model, the statistical analyses of doublets as well as a more in depth description of the PLX diet model. Taken together, your findings elegantly describe a novel dynamics of microglial self-renewal and maturation and the role of a key pathway in this process.

**Decision letter after peer review:**

Thank you for submitting your article "P2Y12 is not a major regulator of the kinetics of microglial self-renewal and maturation in the adult visual cortex" for consideration by *eLife*. Your article has been reviewed by 3 peer reviewers, and the evaluation has been overseen by a Reviewing Editor and Satyajit Rath as the Senior Editor. The following individual involved in review of your submission has agreed to reveal their identity: Amanda Sierra (Reviewer #1).

The reviewers have discussed the reviews with one another and the Reviewing Editor has drafted this decision to help you prepare a revised submission.

All reviewers highlighted the interest of your study that provides an elegant longitudinal study and mathematical modeling of microglial self-renewal and maturation. However, they also raised specific major and major points that should be addressed in a revised version of the manuscript. In particular, they stressed the importance to provide:

i) A clearer description of the mathematical model;

ii) A clarification of the notion of doublets, their statistical analyses as well as an in vivo evidence that they correspond to cell division;

iii) Stronger statistical analyses of the changes in cytokines after PLX diet and of the impact of P2Y12 inactivation on microglial morphology.

We look forward to receiving a revised version of your manuscript and thank you for considering *eLife*.

*Reviewer #1:*

The manuscript "P2Y12 is not a major regulator of the microglial self-renewal and maturation in the adult visual cortex" by Mendes et al. is an elegantly written manuscript that shows very solid data on the kinetics of microglial self-renewal and acquisition of functional maturation after depletion with one of the most promising therapeutic tools in neurodegenerative diseases, the CSf1 inhibitor PLX5266. The authors also show that this process is largely independent of P2Y12, a receptor that has been claimed essential to maintain microglial homeostasis.

The data presented is novel, the methods of analysis are the state-of-the-art and several new tools, including a mathematical model have been developed. There are however, a few issues that need to be addressed by the authors.

1. The mathematical model, while very interesting, is very poorly described for a lay audience. As an author of papers that include mathematical models, I am well aware that this is extremely difficult, but the authors need to be clearer both in the methods and in the Results sections.

In the methods, the authors need to describe the rationale for the different formulas proposed to analyze the dynamics of microglia, and the previous references in which their analysis was surely based.

In the results, the authors need also to be more specific (without entering the formula details). I would suggest 1, the draw a cartoon that summarizes how the model was created; 2, create a table where the values used to create the model (both experimental and simulated); and 3, be more specific about the actual comparison between experimental and simulated data. For instance, I have trouble understanding what the "empirical division rates" are (and how are they calculated), as they first appear in line 353 but they were not mentioned earlier.

2. The authors claim that they did not observe changes in cytokine production after 7 days of the PLX diet, but this is due to the low n of their sampling and the large variability (ouliers?) in the PLX group, as in fact strong tendencies can be observed in most cytokines analyzed in Supp. Figure 2. The authors should perform power analysis to determine the n-size, and reach that n if it is feasible. At the very least, they should be more cautious describing their results and concluding a lack of inflammatory response (line 182).

3. While I agree that visually the P2Y12 KO mice data looks largely like the control mouse data throughout the paper, it is unclear whether in all experiments a proper statistical analysis comparing control and KO mice was performed. For instance, data in Figure 2B vs 2D, 2C vs 2E, 5B vs 5G, and all Figure 6 (D vs G, E vs H, F vs I, K vs M, L vs N).

Certainly, showing each set of data in separate graphs (as is done in most of the paper), does not allow a direct comparison. I understand this is done due to the complexity of the data, but the readers should be better informed on the test that was used to analyze the data.

All figures should include more information on the test used, even when the comparison was not significant.

4. Definition of a doublet: as doublets are a critical point to analyze microglial dynamics in this paper ("The characteristic morphologies of microglia in the process of division allowed us to identify "potential" proliferating cells, which we refer to as doublet somas" (line 269)), the authors should be more precise in their description. Are they simply enlarged microglial cell bodies? How large? In figure 2, their identification would be easier for the reader if the contours of what the authors consider to be the cell bodies are drawn.

Similarly, how can the authors be sure microglia in Figure 3 is a multinucleated body (line 239, pointed with an arrow on the figure), if a nuclei staining was not performed? In this particular cell, the authors claim that it did not divide, but at day 4 there is something that looks like another cell right on top – can the authors confirm this?

5. The authors made some interesting observations on the mode of proliferation of microglia that should be quantified to support their conclusions:

– "The appearance of new cells was associated with a characteristic increase and elongation of the cell soma" (line 257).

– "The original microglia often had extensive processes which were maintained during the division and generation of a new microglia" (line 259).

– "newly generated microglia frequently had a ramified microglia arbor on the same day that division was complete" (line 261).

– "newly generated microglia frequently had a ramified microglia arbor on the same day that division was complete. In many cases, we observed that newly born microglia had a long terminal process extending" (line 263) (this is unclear to me from the image.

– "While one microglia cell remained stable within the network of cells and persisted at the imaged position over time, the other microglia moved away and took up new territory adjacent to the parent cell (line 265)".

The frequency of these events should be quantified.

6. In the discussion, the comparison between the maturation of microglia during development and the maturation of microglia observed in this paper is kind of confusing, mostly because most developmental analysis has been done at the transcriptional level (using scRNAseq), whereas this paper is solely based on imaging. I agree that performing scRNAseq is out of the scope of this paper, but the authors should be more precise in their comparison.

An example of this mix up is the sentence "While microglial expression in the early phase of development appears to recapitulate developmental programs (19), we did not observe the typical morphologies of developing microglia, or changes in process dynamics and responses to injury that could indicate a less mature phenotype." (line 591).

Also, what are the "typical morphologies of developing microglia"? and what is known about the motility of developing microglia?

Finally, the use of the word "profile" (line 593) is confusing, because it is usually related to the "transcriptional profile" but the authors seem to use it to refer to their own morphological results.

7. The paper on Mac2+ (possible) microglia progenitors, which is widely discussed in this paper, has not been peer-reviewed and I am unsure about the policies of *eLife* of citing papers from open repositories such as biorxiv.

*Reviewer #2:*

In this manuscript, the authors have studied the behavior of microglial self-renewal upon PLX-induced depletion. They first performed in vivo imaging in head-fixed animal and thereby confirmed (ref 6 and 9) that microglia displayed limited migration and turnover. Then, they analyzed the dynamics of repopulation after PLX-induced depletion. They confirmed (ref. 18) that the rapid replenishment observed after depletion is only due to residual microglia, and they propose a mathematical model to support this observation. The authors then show that new-born microglia display a ramified morphology, and a dynamic behavior that is comparable to the one observed before depletion. Finally, the authors show that the parameters of replenishment are not altered by the loss of function of P2Y12R.

A major concern is that most of the data in this manuscript simply confirmed already published experiments. This manuscript also provides new data related to brain replenishment after PLX-induced depletion of microglia. These data will certainly be of great interest for those working in the field of microglia function in the adult, but may have a limited impact outside of this community.

*Reviewer #3:*

This is an interesting study on microglia self-renewal and maturation in the adult cortex after microglia depletion. The aims are to understand the mechanisms explaining the rapid repopulation and to test whether new microglia are identical to those which have been depleted. There have already been several studies on this subject. The interest of this new study lies in the fact that the authors performed a longitudinal (over 30 days) analysis of microglia dynamics in awake mice.

The results suggest that the rapid population is based on the local proliferation of microglia that escaped depletion and that the new microglia rapidly acquire functional properties typical of adult microglia. The dynamics of this repopulation is largely independent of P2Y12 receptors.

1. An important part of the article is based on the analysis of microglia called "doublet", which would be microglia in the process of proliferation. However, the authors do not provide a very clear definition of what a doublet is and how these doublets were identified. Clearly defined quantitative criteria would make this part more convincing. In addition, as a certain number of these doublets remain in the doublet state, without dividing, during the observation period (up to 30 days), it would be interesting to show that they indeed correspond to cells in the process of proliferation. A correlation between the identification of doublets in vivo and histochemical analysis of nuclear markers after fixation could help.

2. The approach consisting in using the values of the repopulation dynamics obtained with monitoring microglia every 4 hours for 24 hours in a mathematical model (Figure 4) to test whether proliferation alone can explain the observations made with a monitoring less frequent (1 / day) but longer (30 days) (Figure 2) is interesting. Unfortunately, the description of the mathematical modeling is unclear and it is difficult as it is to understand if the results of this modeling actually support the authors' conclusion that "it is reasonable to believe that the repopulation.… came purely from local doublet cell division" (page 16).

3. The authors tend to minimize the role of P2Y12. It is true that this receptor does not modify the dynamics of microglia repopulation. However, after repopulation the morphology of P2Y12 KO microglia is very different from that of wild type microglia. Suppl. Figure 4 clearly shows that at the end of repopulation (D30), P2Y12 KO microglia look more like the control, wild type and P2Y12 KO, microglia than do repopulating wild type microglia. Maybe this could be stressed a bit more clearly as this may be important for microglia-neuron interactions. As for wild type microglia (Figure 5A), pictures of P2Y12 KO microglia in control and at different stages of the repopulation should be shown.

---

## [Author Response]

All reviewers highlighted the interest of your study that provides an elegant longitudinal study and mathematical modeling of microglial self-renewal and maturation. However, they also raised specific major and major points that should be addressed in a revised version of the manuscript. In particular, they stressed the importance to provide:i) A clearer description of the mathematical model;

To address the comments on the mathematical model in the manuscript, we included a schematic and more clarification on the description of the model in the text accompanying figure 4 and supplementary figure 7.

ii) A clarification of the notion of doublets, their statistical analyses as well as an in vivo evidence that they correspond to cell division;

To address this comment, we quantified the morphology of doublets on a set of confocal images from fixed brains of control animals and animals that were treated with PLX and allowed to repopulate for 2 days (this is peak repopulation time with a large number of doublet somas present) to determine quantitatively the morphological characteristics that separated singlets from doublets. The somata of doublet cells in repopulated animals were enlarged and irregular in size. While no single morphological measurement (soma area, perimeter, elongation; Figure 3) fully separated doublet and singlet somas, a combination of soma size and perimeter separates out the doublet population from non-doublet microglia in repopulated animals, and from microglia in control animals (Figure 3). Doublets are therefore defined as cells with large somas which are elongated, often to form an irregular soma shape. We also immunoreacted these sections and find an upregulation of Galectin-3 in repopulating microglia, although both doublet and singlet microglia showed high levels of expression (Suppl. Figure 6), consistent with results on proliferation and Galectin-3 expression in Zhan et al., 2020.

iii) Stronger statistical analyses of the changes in cytokines after PLX diet and of the impact of P2Y12 inactivation on microglial morphology.

Cytokine analysis: Based on the data presented in old Suppl. Figure 2, we performed a power analysis and determined that a sample size of 7 animals per group allows us to detect a 2-fold increase in cytokine production with a power of 0.8. We then repeated this experiment with a new cohort of animals and present the data in new Suppl. Figure 2. No significant upregulation of any of the cytokines tested was apparent after PLX treatment. P2Y12 comparisons: We now provide the comparisons between P2Y12KO and CX_3_Cr1^GFP/+^ animals in supplementary figures 4-5,9-13.

Reviewer #1:[…] 1. The mathematical model, while very interesting, is very poorly described for a lay audience. As an author of papers that include mathematical models, I am well aware that this is extremely difficult, but the authors need to be clearer both in the methods and in the Results sections.In the methods, the authors need to describe the rationale for the different formulas proposed to analyze the dynamics of microglia, and the previous references in which their analysis was surely based.In the results, the authors need also to be more specific (without entering the formula details). I would suggest 1, the draw a cartoon that summarizes how the model was created; 2, create a table where the values used to create the model (both experimental and simulated); and 3, be more specific about the actual comparison between experimental and simulated data. For instance, I have trouble understanding what the "empirical division rates" are (and how are they calculated), as they first appear in line 353 but they were not mentioned earlier.

We included a schematic that describes the mathematical model in its entirety as supplementary figure 7. In addition, we expanded on the mathematical model in the Results section that accompanies figure 4. We also included a detailed description of the formula used for the model which was created specifically for this analysis by Dr. Matthew McCall and Zachary Brehm, both co-authors on the manuscript.

2. The authors claim that they did not observe changes in cytokine production after 7 days of the PLX diet, but this is due to the low n of their sampling and the large variability (ouliers?) in the PLX group, as in fact strong tendencies can be observed in most cytokines analyzed in Supp. Figure 2. The authors should perform power analysis to determine the n-size, and reach that n if it is feasible. At the very least, they should be more cautious describing their results and concluding a lack of inflammatory response (line 182).

To address this question, we performed a power analysis in consultation with Dr. Matthew McCall. We found that 7 animals for each group (control versus PLX) were necessary to achieve a power of 0.8 to detect a 2-fold change in cytokine release. We decided on a conservative 2-fold changes although most papers using inflammatory stimuli show upregulations of cytokines that are at least several fold and in some cases an order of magnitude. Using a new cohort of animals, we investigated how PLX treatment affected IL-1b, IL-4, IL-6, IL-10 and TNF-a and found that there was no observed change in the cytokine production after PLX (supplementary figure 2). This is in line with previously published work using PLX5622.

3. While I agree that visually the P2Y12 KO mice data looks largely like the control mouse data throughout the paper, it is unclear whether in all experiments a proper statistical analysis comparing control and KO mice was performed. For instance, data in Figure 2B vs 2D, 2C vs 2E, 5B vs 5G, and all Figure 6 (D vs G, E vs H, F vs I, K vs M, L vs N).Certainly, showing each set of data in separate graphs (as is done in most of the paper), does not allow a direct comparison. I understand this is done due to the complexity of the data, but the readers should be better informed on the test that was used to analyze the data.All figures should include more information on the test used, even when the comparison was not significant.

We provided the comparisons between P2Y12KO and CX_3_Cr1^GFP/+^ animals in supplementary figures 4-5,9-13.

4. Definition of a doublet: as doublets are a critical point to analyze microglial dynamics in this paper ("The characteristic morphologies of microglia in the process of division allowed us to identify "potential" proliferating cells, which we refer to as doublet somas" (line 269)), the authors should be more precise in their description. Are they simply enlarged microglial cell bodies? How large? In figure 2, their identification would be easier for the reader if the contours of what the authors consider to be the cell bodies are drawn.

To address this comment, we quantified the morphology of doublets on a set of confocal images from fixed brains of control animals and animals that were treated with PLX and allowed to repopulate for 2 days (this is peak repopulation time with a large number of doublet somas present) to determine the morphological characteristics that separated singlets from doublets. The somata of doublet cells in repopulated animals were enlarged and irregular in size. While no single morphological measurement (soma area, perimeter, elongation; Figure 3) fully separated doublet and singlet somas, a combination of soma size and perimeter largely separates out the doublet population from non-doublet microglia in repopulated animals, and from microglia in control animals (Figure 3). Doublets are therefore defined as cells with large somas (generally over 100um^2^ in cross section, compared to an average of 75um^2^ for control cells) which are elongated, often to form an irregular soma shape.

Similarly, how can the authors be sure microglia in Figure 3 is a multinucleated body (line 239, pointed with an arrow on the figure), if a nuclei staining was not performed? In this particular cell, the authors claim that it did not divide, but at day 4 there is something that looks like another cell right on top – can the authors confirm this?

We only observed a large mass of microglia that may represent a multinucleated body once in all our imaging sessions and thus were not able to analyze this phenomenon. While multinucleated bodies have been observed in the Bruttger et al., 2015 paper, we do not believe they contribute to microglial proliferation after depletion with PLX5662.

5. The authors made some interesting observations on the mode of proliferation of microglia that should be quantified to support their conclusions:

We quantified these observations in a subset of doublets (22 from 3 animals) that could be clearly followed across imaging time points with high spatial and temporal resolution (from the imaging interval of 4 hours shown in Figure 4).

– "The appearance of new cells was associated with a characteristic increase and elongation of the cell soma" (line 257).

91% of new cells were associated with a characteristic increase and elongation of the cell soma of the original cell.

– "The original microglia often had extensive processes which were maintained during the division and generation of a new microglia" (line 259).

95% of the original microglia had extensive process which were maintained during the division and generation of a new microglia.

– "newly generated microglia frequently had a ramified microglia arbor on the same day that division was complete" (line 261).

82% newly-generated microglia had a ramified microglia arbor on the same day that division was complete.

– "newly generated microglia frequently had a ramified microglia arbor on the same day that division was complete. In many cases, we observed that newly born microglia had a long terminal process extending" (line 263) (this is unclear to me from the image.

We removed this observation as we found it difficult to provide an objective definition of this phenomenon.

– "While one microglia cell remained stable within the network of cells and persisted at the imaged position over time, the other microglia moved away and took up new territory adjacent to the parent cell (line 265)".

We modified the language of this in the text to reflect that it is difficult to identify and differentiate between parent cells and new cells.

This data was included in the text in the Section “Residual microglia are capable of rapid division to generate new microglia and repopulate the cortex”.

6. In the discussion, the comparison between the maturation of microglia during development and the maturation of microglia observed in this paper is kind of confusing, mostly because most developmental analysis has been done at the transcriptional level (using scRNAseq), whereas this paper is solely based on imaging. I agree that performing scRNAseq is out of the scope of this paper, but the authors should be more precise in their comparison.An example of this mix up is the sentence "While microglial expression in the early phase of development appears to recapitulate developmental programs (19), we did not observe the typical morphologies of developing microglia, or changes in process dynamics and responses to injury that could indicate a less mature phenotype." (line 591).

We revised this section and clarified the maturation of microglia in development versus the maturation of microglia with depletion and repopulation.

Also, what are the "typical morphologies of developing microglia"? and what is known about the motility of developing microglia?

We have explained the morphological and dynamic changes in microglia during development (ameboid morphologies and motile somas vs. processes in early life microglia) to compare to our findings in adult newly-born microglia.

Finally, the use of the word "profile" (line 593) is confusing, because it is usually related to the "transcriptional profile" but the authors seem to use it to refer to their own morphological results.

We removed the word profile when referring to the morphological results throughout the manuscript.

7. The paper on Mac2+ (possible) microglia progenitors, which is widely discussed in this paper, has not been peer-reviewed and I am unsure about the policies of eLife of citing papers from open repositories such as biorxiv.

We updated the references to include the published work.

Reviewer #2:[…] A major concern is that most of the data in this manuscript simply confirmed already published experiments. This manuscript also provides new data related to brain replenishment after PLX-induced depletion of microglia. These data will certainly be of great interest for those working in the field of microglia function in the adult, but may have a limited impact outside of this community.

While we agree that some of our data confirms recently published findings, we are confident that the novel aspects of our experiments will not only add to the microglia repopulation field but to the microglia biology field altogether. We provide new information as to the dynamics of microglial division, showing that these divisions can be remarkably rapid and continuous. We also provide novel information on the maturation of newly generated microglia and show that their morphologies, surveillance and responses to injury are established very rapidly after repopulation. Lastly, we show that P2Y12 has a limited role in these processes, despite being a critical receptor for microglial homeostatic functions. An overall understanding of the basic biology and mechanism behind adult microglia biology will be beneficial in understanding how microglia self-renew in disease states.

Reviewer #3:[…] 1. An important part of the article is based on the analysis of microglia called "doublet", which would be microglia in the process of proliferation. However, the authors do not provide a very clear definition of what a doublet is and how these doublets were identified. Clearly defined quantitative criteria would make this part more convincing. In addition, as a certain number of these doublets remain in the doublet state, without dividing, during the observation period (up to 30 days), it would be interesting to show that they indeed correspond to cells in the process of proliferation. A correlation between the identification of doublets in vivo and histochemical analysis of nuclear markers after fixation could help.

To address this comment, we stained and characterized doublets during the peak of division on day 2 of repopulation and now provide a quantitative analysis of the morphological features that distinguish doublets from singlets and control microglia (Figure 3). Additionally, we also stained these cells for Galectin 3, to determine whether the doublet morphology corresponded to the recently described Gal3 positive progenitor-like microglia that is resistant to CSF1R inhibition (Zhan et al. 2020). While we confirmed that Gal3 was indeed upregulated in a subset of microglia during the repopulation phase, it appeared that Gal3 was expressed equally in doublet and singlet cells (see new Supplementary Figure 6). This agrees with the findings in the Zhan study that described EdU incorporation equally in Gal3+ and Gal3- microglia during repopulation, suggesting that proliferation occurs in both of these types of microglia. Unfortunately, Ki67 labels very few microglia in our hands in repopulated tissue (and in the Zhan et al., 2020 – see Figure 5 supplement 1) precluding an analysis of proliferating cells and morphology. We did not attempt to use EdU as the experimental design requires a pulse/chase administration which would label many cells rather than just those actively dividing at the time of analysis.

2. The approach consisting in using the values of the repopulation dynamics obtained with monitoring microglia every 4 hours for 24 hours in a mathematical model (Figure 4) to test whether proliferation alone can explain the observations made with a monitoring less frequent (1 / day) but longer (30 days) (Figure 2) is interesting. Unfortunately, the description of the mathematical modeling is unclear and it is difficult as it is to understand if the results of this modeling actually support the authors' conclusion that "it is reasonable to believe that the repopulation.… came purely from local doublet cell division" (page 16).

We included a schematic that describes the mathematical model in its entirety in the supplementary figures section. In addition, we expanded on the mathematical section that accompanies figure 5 We also included a detailed description of the formula used for the model. This description was created and reviewed by statisticians Dr. Matthew McCall and Zachary Brehm both co-authors on the manuscript.

3. The authors tend to minimize the role of P2Y12. It is true that this receptor does not modify the dynamics of microglia repopulation. However, after repopulation the morphology of P2Y12 KO microglia is very different from that of wild type microglia. Suppl. Figure 4 clearly shows that at the end of repopulation (D30), P2Y12 KO microglia look more like the control, wild type and P2Y12 KO, microglia than do repopulating wild type microglia. Maybe this could be stressed a bit more clearly as this may be important for microglia-neuron interactions. As for wild type microglia (Figure 5A), pictures of P2Y12 KO microglia in control and at different stages of the repopulation should be shown.

Thank you for your comment. It is true that P2Y12 may play a role in the regulation of newly born microglia morphology. To address this comment, we altered the title of the paper, rewrote the abstract and discussion to reflect this point and included pictures of P2Y12 KO microglia in figure 6. Direct comparisons of CX3cr1 GFP/+ and P2Y12 KO animals are now presented in supplementary figures 4-5 and 9-13.

References:

1. Lowery, R.L., Tremblay, M.E., Hopkins, B.E., and Majewska, A.K. (2017). The microglial fractalkine receptor is not required for activity- dependent plasticity in the mouse visual system. Glia *65*, 1744-1761. 10.1002/glia.23192.

2. Sipe, G.O.e.a. (2016). Microglial P2Y12 is necessary for synaptic plasticity in mouse visual cortex. Nat Commun *7*, 10905. 10.1038/ncomms10905.

3. Stowell, R.D., Sipe, G.O., Dawes, R.P., Batchelor, H.N., Lordy, K.A., Whitelaw, B.S., Stoessel, M.B., Bidlack, J.M., Brown, E., Sur, M., and Majewska, A.K. (2019). Noradrenergic signaling in the wakeful state inhibits microglial surveillance and synaptic plasticity in the mouse visual cortex. Nat Neurosci *22*, 1782-1792.

10.1038/s41593-019-0514-0.